# An epigenetic switch ensures transposon repression upon dynamic loss of DNA methylation in embryonic stem cells

**Marius Walter[1,2,3,4], Aurélie Teissandier[1,3,4,2,5,6,7], Raquel Pérez-Palacios[1,3,4,2], Déborah Bourc'his[1,3,4,2]\***

[1]Department of Genetics and Developmental Biology, Institut Curie, Paris, France; [2]Paris Science Lettres Research University; [3]UMR3215, CNRS, Paris, France; [4]U934, Inserm, Paris, France; [5]Bioinformatics, Biostatistics, Epidemiology and Computational Systems Biology of Cancer, Institut Curie, Paris, France; [6]Mines Paris Tech, Paris, France; [7]U900, Inserm, Paris, France

**Abstract** DNA methylation is extensively remodeled during mammalian gametogenesis and embryogenesis. Most transposons become hypomethylated, raising the question of their regulation in the absence of DNA methylation. To reproduce a rapid and extensive demethylation, we subjected mouse ES cells to chemically defined hypomethylating culture conditions. Surprisingly, we observed two phases of transposon regulation. After an initial burst of de-repression, various transposon families were efficiently re-silenced. This was accompanied by a reconfiguration of the repressive chromatin landscape: while H3K9me3 was stable, H3K9me2 globally disappeared and H3K27me3 accumulated at transposons. Interestingly, we observed that H3K9me3 and H3K27me3 occupy different transposon families or different territories within the same family, defining three functional categories of adaptive chromatin responses to DNA methylation loss. Our work highlights that H3K9me3 and, most importantly, polycomb-mediated H3K27me3 chromatin pathways can secure the control of a large spectrum of transposons in periods of intense DNA methylation change, ensuring longstanding genome stability.

*For correspondence: deborah. bourchis@curie.fr

**Competing interests:** The authors declare that no competing interests exist.

## Introduction

Millions of transposable elements reside in mammalian genomes, far surpassing in number the approximately 25000 protein-coding genes (*Lander et al., 2001*). Most of these elements are retro-transposons, which utilize an RNA intermediate to duplicate and mobilize. Through their activity or their mere presence, transposons can be both beneficial for the evolution of the host genome and deleterious for its integrity. They can modify gene functions through insertional mutagenesis, influence gene transcriptional outputs by acting as promoters or enhancers or induce chromosomal rearrangements through non-allelic recombination (*Goodier and Kazazian, 2008*). Accordingly, erratic transposon-related events have been linked to congenital diseases, cancer and infertility (*Kaer and Speek, 2013*).

Successive waves of transposon expansion and decline have shaped mammalian genomes over evolution, leading to a current occupancy rate of approximately half of the genomic space. Reflecting their various evolutionary origin and multiplication success, resident elements are greatly diverse in structures, numbers and functional properties, which define discrete families of transposons. Long Terminal Repeat (LTR) sequences characterize endogenous retroviruses (ERVs, 12% of the mouse genome), which can be further subdivided into three families (ERV1, ERVK and ERVL), according to the infectious retroviruses they derive from (*Stocking and Kozak, 2008*). Non-LTR elements

**eLife digest** Transposons are sequences of DNA with the ability to mobilize and jump from one position to another. In the human genome, the number of transposons far surpasses the number of genes. Furthermore, while transposons have been beneficial for the evolution of the human genome, they can also alter genes and cause cancer and genetic diseases.

The danger posed by transposons has led to numerous mechanisms to keep them under control. In particular, a natural biochemical modification of the DNA molecule called "DNA methylation" plays an important role in keeping transposons inactive or silent. However, during the early development of an embryo, the DNA methylation marks are erased throughout the entire genome. This provides an opportunity for transposons to be active, and it is not clear how the genome manages to control transposons in the absence of this essential protective mark.

Walter et al. have now investigated this process by using a cell type that mimics the loss of DNA methylation that occurs during embryonic development – mouse embryonic stem cells grown in the laboratory. The experiments revealed that when DNA methylation is lost progressively, the transposons are reactivated at first but are later put back into a silent mode by alternative mechanisms. These mechanisms compensate for the disappearance of DNA methylation by encouraging the DNA around transposons to become compacted, which prevents the transposons from moving.

Further analysis revealed that the different families of transposons that exist in the mouse genome can be classified into three groups, and in each group different proteins ensure the transposons remain repressed in the absence of DNA methylation. Together these findings reveal that multiple pathways cooperate to protect the genome against the activity of a variety of transposons. Finally, in mammals, DNA methylation is naturally erased both during the formation of sperm and egg cells and in the early embryo. As such, it will be important to verify whether the mechanisms discovered in the laboratory-grown cells also tame transposons during these critical developmental periods.

comprise Long and Short INterspersed Elements (LINEs and SINEs, 20% and 8% of the genome, respectively), and also consist of specific sub-families (*Babushok et al., 2007*). The majority of transposons have accumulated nullifying mutations and truncations, but around 1–2% of LINEs and ERVs have intact sequences that embed the protein coding information necessary for their mobilization. Notably, ERVK elements show the greatest level of activity, which causes at least 10% of spontaneous mutations in laboratory mice (*Maksakova et al., 2006*).

To minimize their impact on genome fitness, multiple layers of control antagonize transposons at different steps of their life cycle (*Zamudio and Bourc'his, 2010*). Notably, restraining mechanisms can differ between cell types. In somatic cells and in the male differentiating germline, DNA methylation is the main transcriptional suppressor of LTR and non-LTR transposons. In these contexts, transposable elements are densely methylated (*Rollins et al., 2006*; *Smith et al., 2012*) and DNA hypomethylation leads to their de-repression (*Bourc'his and Bestor, 2004*; *Walsh et al., 1998*). In contrast, the early germline and the early embryo manage to globally control their transposon burden without DNA methylation. These cells naturally undergo genome-wide loss of DNA methylation, likely as part of the acquisition of a pluripotent, flexible state (*Seisenberger et al., 2013*). Moreover, genetic studies have demonstrated that mouse embryonic stem (ES) cells can use DNA methylation-independent mechanisms to silence transposons: knocking-out the three active DNA methyltransferases (*Dnmt*-tKO) does not yield significant de-repression of transposons, except Intracisternal A Particle (IAP) elements (*Karimi et al., 2011b*; *Matsui et al., 2010*)

In fact, transposon control in ES cells seems to rely primarily on post-translational histone methylation, notably at lysine 9 of histone H3 (H3K9). H3K9 dimethylation (H3K9me2), which is deposited by the EHMT2/G9a and EHMT1/GLP lysine methyltransferases, directly and specifically represses class L ERVs (*Maksakova et al., 2013*). H3K9 trimethylation (H3K9me3) can be catalyzed by the SETDB1 (also known as ESET) or the SUV39H enzymes. The SUV39H system targets H3K9me3 at evolutionary young LTR and non-LTR transposons, but *Suv39h* mutant ES cells principally up-regulate

LINE1 elements (*Bulut-Karslioglu et al., 2014*). In parallel, SETDB1, together with its associated co-repressor, the Krüppel-associated box domain (KRAB)-Associated Protein 1 (TRIM28, also known as KAP1), mainly control H3K9me3-dependent suppression of ERVK transposons- a family to which IAP elements belong (*Karimi et al., 2011b*; *Matsui et al., 2010*; *Rowe et al., 2010*). TRIM28 is recruited to specific genomic sites via direct interactions with KRAB-zinc finger proteins (*Friedman et al., 1996*), which are a large family of DNA binding factors that co-evolved with ERVs (*Emerson and Thomas, 2009*). Therefore, different H3K9 methylation-based mechanisms are utilized to silence different transposons families in ES cells. In contrast, the repressive spectrum of polycomb-mediated H3 lysine 27 trimethylation (H3K27me3) is limited: only Murine Leukemia Virus (MuLV) elements are reactivated upon H3K27me3 deficiency (*Leeb et al., 2010*).

However, the prevailing view that H3K9 methylation acts as the main transposon controller in ES cells may be biased by two confounding factors. First, conclusions are based on analyses of chromatin modifier mutants, which still harbor high DNA methylation levels. Second, proper transposon repression in *Dnmt*-tKO ES cells may reflect a long-term adaptation to a DNA methylation-free state rather than a lack of significant role of DNA methylation per se. In fact, how the ES cell genome transitions from a DNA methylation-dependent to -independent mode of transposon control has never been investigated.

To study the dynamics of transposon regulation upon DNA methylation loss, we modulated the ES cell methylome by using interconvertible culture systems, which do not modify pluripotency potential. ES cells grown in standard serum-based conditions have heavily methylated genomes (~75% of CpG methylation)(*Stadler et al., 2011*), which is linked to the expression of *de novo* DNA methyltransferases. ES cells grown in presence of two small kinase inhibitors (2i) down-regulate these enzymes, and have reduced DNA methylation levels (*Leitch et al., 2013*; *Ying et al., 2008*). Upon transfer from serum to 2i medium, demethylation occurs with a slow kinetics: several weeks are required to reach 20–30% of CpG methylation. Notably, imprinted genes, major satellite repeats and IAP elements maintain persistent DNA methylation after 2i adaptation (*Ficz et al., 2013*; *Habibi et al., 2013*). Addition of vitamin C (vitC) can also lower the ES cell methylome. This compound promotes active demethylation by stimulating the TET (Ten Eleven Translocation) enzymes, which oxidize 5-methylcytosines to 5-hydroxymethylcytosines that are potential intermediates towards unmethylated cytosines (*Blaschke et al., 2013*).

Here, by switching ES cells directly from a serum-based to a 2i+vitC medium, we were able to induce rapid and extensive demethylation genome-wide, mimicking a situation occurring in the early embryo. By combining DNA methylation, chromatin and transcriptional profiling of transposons along with genetic analyses, we found that DNA methylation represses multiple families of transposons in ES cells, but an epigenetic switch towards histone-based control is progressively implemented as DNA methylation disappears. Importantly, we reveal for the first time the specific and overlapping roles of H3K9 and H3K27 trimethylation in controlling distinct transposon families upon DNA demethylation. These findings have important implications for understanding the molecular underpinning of transposon control in the pluripotent cells of the early mammalian embryo.

## Results

### DNA methylation is rapidly and extensively lost in ES cells during serum to 2i+vitC media conversion

*Dnmt*-tKO ES cells are completely devoid of DNA methylation, yet expression levels of most transposable elements remain globally similar to wild-type (WT) ES cells (*Karimi et al., 2011b*; *Tsumura et al., 2006*). This may indicate the implementation of alternative mechanisms that compensate for DNA methylation-based repression. To analyze dynamic adaptation, we utilized a culture-based system that results in rapid DNA methylation loss: converting ES cells from serum-based to 2i+vitamin C (2i+vitC) culture conditions. To overcome confounding genetic effects, we used the J1 ES cell line, from which *Dnmt*-tKO mutants were originally derived.

Quantification using the methyl-CpG sensitive restriction enzyme-based LUminometric Methylation Assay (LUMA) (*Karimi et al., 2011a*) revealed that CpG methylation linearly decreased from 77% to 13% in six days, and reached a minimal level of 6% after 14 days of conversion (*Figure 1A*). In comparison, cells grown in serum+vitC or in 2i-only maintained relatively high CpG methylation

content after the same treatment duration, with an average of 56% and 22%, respectively (*Figure 1—figure supplement 1A*), in agreement with a previous study (*Habibi et al., 2013*). This suggests that such a rapid and extensive loss of genomic methylation can only be attained through the synergistic action of 2i-dependent passive demethylation and vitC-dependent active demethylation.

To monitor the demethylation dynamics of specific genomic sequences, we performed quantitative bisulfite-pyrosequencing (*Figure 1B*). All analyzed sequences reached very low levels of CpG methylation upon 2+vitC switch, although at various rates. Young LINE1 transposons (L1-A and L1-T) mirrored the dynamics of the genome average, while the CpG-rich promoter of the germline-specific gene *Dazl* was a fast 'loser', comparatively. Consistent with their intrinsic ability to maintain high levels of DNA methylation in various contexts of global DNA hypomethylation (*Ficz et al., 2013*; *Seisenberger et al., 2013*), the demethylation rate of IAP transposons and the Imprinting Control Region (ICR) of the *H19-Igf2* locus was slower than the rest of genome. Nevertheless, the combination of 2i and vitC eventually overcame chromatin environments that confer protection of these sequences from DNA demethylation.

To determine the extent of DNA demethylation globally in 2i+vitC culture conditions, we carried out whole-genome bisulfite sequencing (WGBS) at the conversion end-point. Quality control indicated high genomic coverage, with approximately 55% of CpGs covered at least five times (*Supplementary file 1*). Available methylome maps indicate that 71% and 30% of CpG sites are methylated in serum and in 2i-only conditions, respectively (*Habibi et al., 2013*; *Seisenberger et al., 2012*; *Figure 1C*, *Figure 1—figure supplement 1B* and *Supplementary file 1*); in contrast, ES cells grown in 2i+vitC were almost completely unmethylated, with an average CpG methylation of 4.6%, which is fully consistent with the LUMA quantification (*Figure 1C*). Low methylation levels were homogeneously found throughout all genomic compartments, including single-copy genic regions and repeated sequences (*Figure 1D* and *Figure 1—figure supplement 1C*). In particular, all transposable element families (ERVs, LINEs and SINEs) were affected by 2i+vitC-induced demethylation (*Figure 1E*). In an attempt to identify individual genomic regions with significant DNA methylation traces (*Song et al., 2013*), we uncovered 4,100 Residually Methylated Regions (RMRs) (*Figure 1F,G*), which exhibited an average of 26% of CpG methylation after long-term 2i+vitC conversion. These were also regions prone to high DNA methylation retention in 2i-only conditions (65% of CpG methylation). We estimated that nearly 75% of the RMRs overlapped with repeated sequences, among which half belonged to the ERVK class. This confirmed the specific ability of these elements, which includes IAPs, to resist genome-wide erasure of DNA methylation. One quarter of the repeat-associated RMRs overlapped with LINEs, however specifically localized around 5' UTR regions; in contrast, ERVK-associated RMRs encompassed the entire length of these elements (*Figure 1G*). Notably, Imprinting Control Regions (ICRs), which are usually protected against DNA methylation erasure in 2i conditions, were devoid of any residual DNA methylation in 2i+vitC (*Figure 1—figure supplements 1D,E*).

Our analyses show that only scarce genomic regions retain DNA methylation in 2i+VitC, and even those regions are lowly methylated when compared to other culture systems. Global CpG methylation levels, less than 5%, are unprecedented in male WT cells, both in culture and in vivo (*Seisenberger et al., 2013*). This experimental system provides a valuable means to study the dynamic adaptation of the genome to a loss of DNA methylation.

## Transposons undergo a biphasic mode of regulation upon serum to 2i +vitC conversion

Using the serum to 2i+vitC medium conversion system, we investigated how transposable elements transcriptionally respond to an acute loss of DNA methylation. Through a time-course RT-qPCR analysis of steady-state levels of three classes of retrotranscripts (LINE1, IAPEz and MERVL), we observed a two-phase pattern: *1)* an initial up-regulation, which culminates at day 6 (D6) of 2i+vitC conversion, when genomic methylation reaches a low plateau, then *2)* re-silencing in the absence of DNA methylation (*Figure 2A*). This was confirmed by amplification-free Nanostring nCounter quantification and further extended to VL30 elements (*Figure 2—figure supplement 1A*). To rule out background-specific effects, we exposed serum-cultured E14 ES cells to 2i+vitC (*Figure 2—figure supplement 1B*). Despite some differences in the magnitude of transposon de-repression observed between the J1 and E14 cell lines, the same biphasic pattern of regulation was reproduced. By contrast, the quantity of transposon transcripts remained constant during the conversion from serum to 2i-only or from

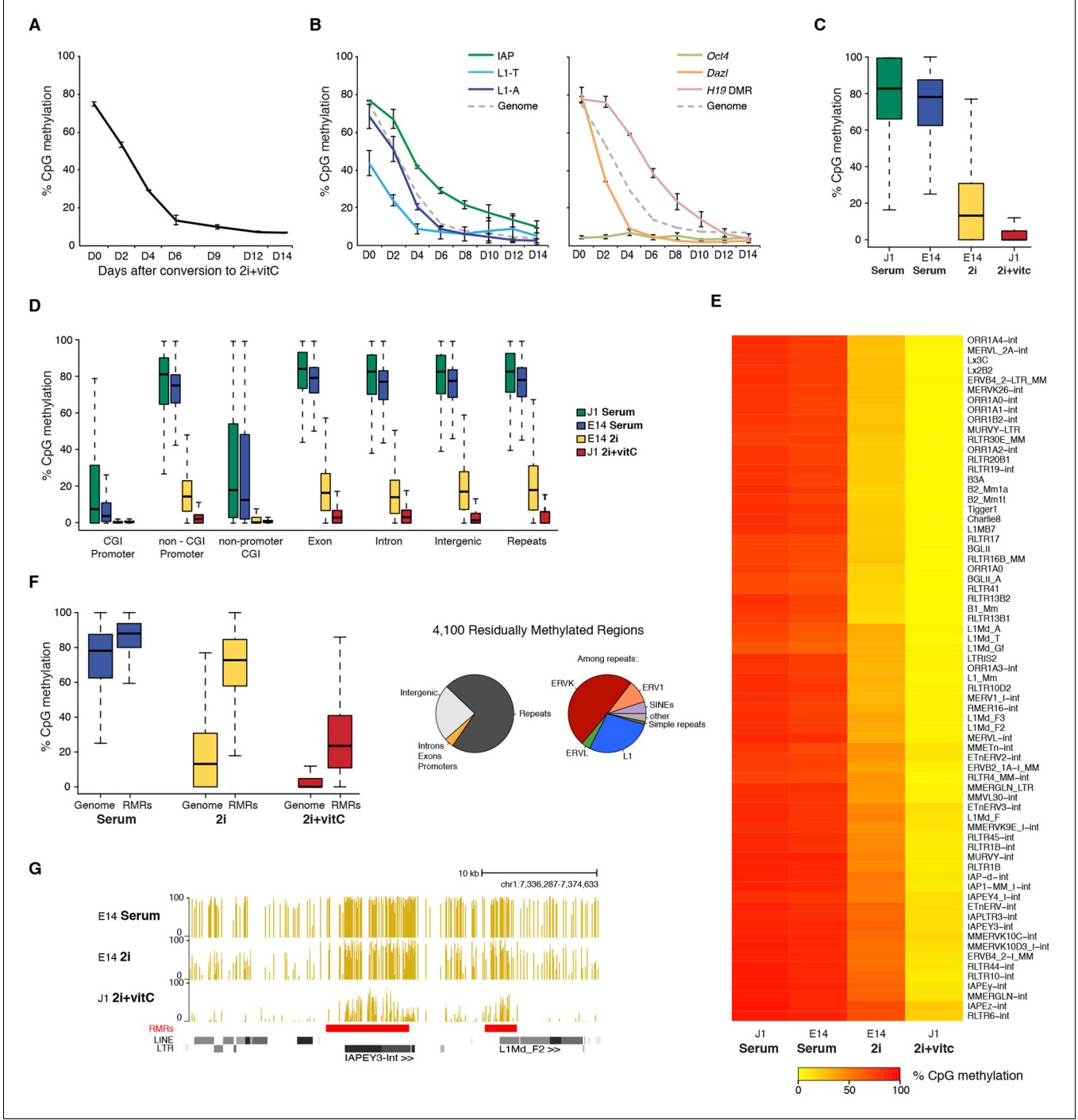

**Figure 1.** Kinetics and extent of DNA methylation loss in ES cells upon serum to 2i+vitC conversion (**A**) Time course of global CpG methylation loss measured by LUMA over 14 days (D0 to D14) of conversion from serum to 2i+vitC. Data represent mean and Standard Error of the Mean (SEM) between two biological replicates. (**B**) Sequence-specific CpG methylation level measured by bisulfite pyrosequencing. Data represent mean ± SEM between two biological replicates. (**C**) Tukey boxplot representation of genome-wide CpG methylation content as measured by WGBS in different culture conditions. Datasets of J1 Serum, E14 Serum and E14 2i were obtained from previous studies (*Habibi et al., 2013*; *Seisenberger et al., 2012*) (**D**) CpG methylation distribution over different genomic compartments by WGBS. (**E**) Heatmap and hierarchical clustering of average CpG methylation over 69 transposon families as measured by WGBS. (**F**) Left panel: Tukey boxplot representation of CpG methylation content in Residually Methylated Regions (RMRs) (*n* = 4,100) compared to the whole genome in various culture conditions. Right panel: pie chart distribution of 2i+vitC RMRs in different genomic compartments (left) and among repeats (right). (**G**) Example of WGBS profile of a genomic region containing two 2i+vitC RMRs mapping to an

*Figure 1 continued on next page*

*Figure 1 continued*

IAPEY and a L1 elements. Bars represent the methylation percentage of individual CpG sites, between 0 (unmethylated) and 100% (fully methylated). Location of LINE and LTR transposons (RepeatMasker) are displayed below; the RMRs are highlighted in red.

The following figure supplement is available for figure 1:

**Figure supplement 1.** DNA methylation is almost completely erased in 2i+vitC medium.

serum to serum+vitC (*Figure 2—figure supplement 1A*), which further underscores the synergistic effect of 2i and vitC in releasing DNA methylation-based repression of transposons. Importantly, the transposon transcription burst did not occur upon conversion of *Dnmt*-tKO cells (*Figure 2—figure supplement 1C*). Rapid transition from a methylated to an unmethylated state seems to provide a window for transposon reactivation; this is in agreement with the hypothesis that *Dnmt*-tKO ES cells have likely acquired long-term compensatory mechanisms preventing this relaxation.

We further found that the burst of transposon expression also occurs at the protein level: both LINE1-ORF1 and IAP-gag proteins presented a peak of expression at D6, which we detected by western blotting (*Figure 2B*) and by quantification of immunofluorescence signals (*Figure 2C* and *Figure 2—figure supplement 2A,B*). While IAP-gag staining was uniform among cells at a given time point, LINE1 protein intensity showed great inter-cellular variability, ranging from intense to no signal. Importantly, the level of LINE1 heterogeneity was present throughout the conversion process from serum to 2i+vitC conditions, including at D6. In an attempt to link this heterogeneity with fluctuating levels of pluripotency, we performed co-staining against NANOG (*Figure 2—figure supplement 2A*). We could not detect any correlations suggesting that LINE1 heterogeneous expression is not linked to various degrees of pluripotency. Additionally, co-staining with phosphorylated-H2AX did not reveal a correlation between the level of transposon expression and DNA damage (data not shown).

We wanted to rule out that the repression phase we observed was not simply a reflection of positive selection of a subset of cells that maintained transposon repression throughout the medium conversion. We found cell proliferation to remain globally constant over the 14-day period of media conversion, as measured by division rate (*Figure 2—figure supplement 3A*), transcriptional level of different proliferation markers (*Figure 2—figure supplement 3B*) or percentage of histone H3 Serine 10 phosphorylation-positive cells (*Figure 2—figure supplement 3C*). Similarly, we did not observe increased cell death/apoptosis at any days during the conversion (*Figure 2—figure supplement 3C*). Finally, despite the transient release of transposon silencing at D6, we failed to detect transposon multiplication or transposon-induced chromosome rearrangements: genomic copy numbers of LINE1 and IAPEz elements as well as karyotypes were globally similar between cells before (D0) and after the transposon burst (D14) (*Figure 2—figure supplement 3D,E*). In sum, 2i+vitC induces a transient up-regulation of transposon transcription and translation, but cellular viability and genome integrity remain largely intact.

## Transposon silencing release occurs at the familial and individual level

To gain a qualitative and quantitative view of the transcriptional dynamics of transposons upon acute loss of DNA methylation, we performed paired-end RNA-seq at D0, D6 and D13 of serum to 2i+vitC conversion, in biological replicates. Typically, to map transposons, the choice is either to allow multiple hits at the expense of specificity, or to consider unique reads only and lose substantial information. Here, we combined the two methods, in order to provide in-depth characterization of the dynamics of transposon regulation at the familial level, while bringing insights into intra-familial heterogeneity. We further improved transposon mapping by correcting the RepeatMasker annotation (*Figure 2—figure supplement 4A*), which tends to overestimate the number of transposon entities by counting a unique element as several individual fragments. This is systematic for ERVs, which are split into internal and LTR sequences, but can also concern any type of transposons with small internal deletions or insertions. Using bioinformatic resources allowing the assembly of different fragments of an element (*Bailly-Bechet et al., 2014*), our reconstructed version gave a census of 588,739 LINEs and 497,706 ERVs, while the original annotation roughly doubles these numbers, with 989,411 LINEs and 969,096 ERVs. Finally, we assigned an integrity score to each element (1 being

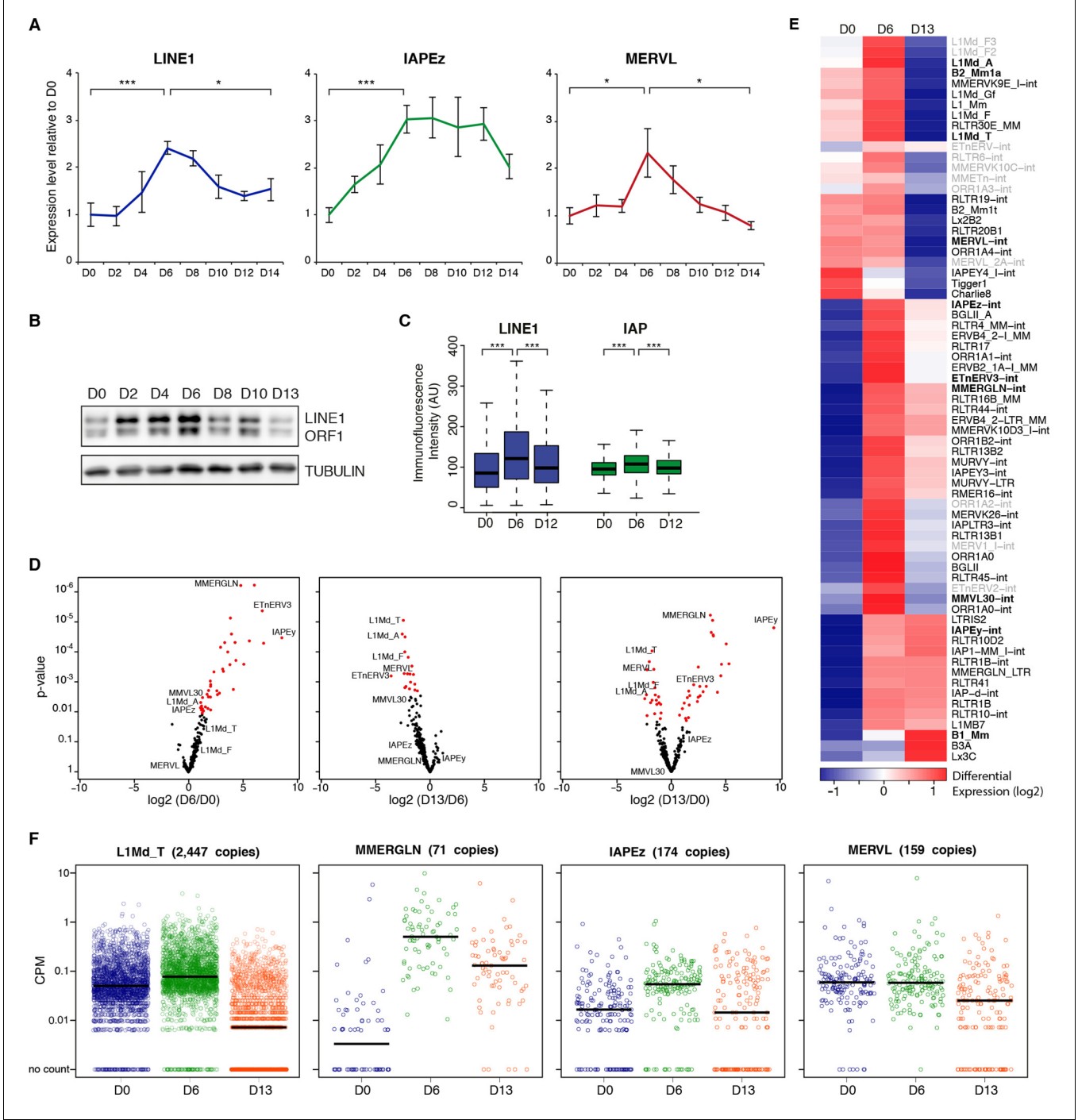

**Figure 2.** Two phases of transposon regulation upon serum to 2i+vitc conversion. (**A**) Dynamic expression of LINE1, IAPEz and MERVL families upon conversion from serum to 2i+vitC as measured by RT-qPCR. Values were normalized to *Gapdh* and *Rplp0* and are expressed as the fold change to D0. Data represent mean ± SEM from five biological replicates. *p<0.05, **p<0.01 and ***p<0.001 (Student's t-test). (**B**) Evolution of LINE1-ORF1 protein levels at different time points during medium conversion. (**C**) Distribution of LINE1-ORF1 and IAP-gag protein levels after ImageJ quantification of immunofluorescence intensity in individual cells. Between 1000 and 5000 cells were analyzed per sample. ***p<0.001 (Wilcoxon rank-sum test) (**D**) Volcano plot representation of up- and down-regulated transposons as measured by RNA-seq between D0 and D6 (left), D6 and D13 (middle), and D0 and D13 (right). Red dots indicate significantly misregulated repeats between two conditions (fold change >2 and p-value<0.05). RNA-seq mapping allowed multiple hits onto the genome. (**E**) Heatmap representation and hierarchical clustering of expression changes for 69 transposon families at D0, D6 and D13. Bold names: transposons of specific interest; grey names: transposons that are not significantly up- or down-regulated between any time points. Colors represent on a log2-scale the differential expression between a given time point and the average of the three time points. (**F**) Expression

*Figure 2 continued on next page*

*Figure 2 continued*

of individual elements from different transposon families at D0, D6 and D13 in Count per Millions (CPM). Each dot represents a single element. RNA-seq mapping allowed only unique hits in the reference genome; only elements with a minimum of 10 reads in at least one of the sample were conserved. The black bar represents the median of the distribution. Analyzed numbers of distinct transposon copies per family appear into brackets.

The following figure supplements are available for figure 2:

**Figure supplement 1.** Two phases of transposon regulation correlate with rapid DNA methylation loss.

**Figure supplement 2.** Cellular heterogeneity of transposon expression.

**Figure supplement 3.** Cell proliferation, cell death, and genome integrity.

**Figure supplement 4.** Genome-wide characterization of transposon relaxation.

the maximum), taking into account deletions, insertions and the divergence rate from the consensus sequence. Using a cutoff of 0.8, we predicted a number of 37,194 relatively intact LINEs (6.3% of total LINE elements) and 15,604 ERVs (3.1%) in the mouse reference genome.

Quality control of our RNA-seq datasets indicated high genomic coverage (*Supplementary file 1*) and consistency between replicates (*Figure 2—figure supplement 4B*). Notably, by excluding transposon-derived reads mapping to RefSeq exons, only autonomously transcribed transposons were considered for this analysis. By allowing multiple hits and by weighting each read by its hit number, a total of 58 transposon families were found differentially expressed between at least two of the time points of medium conversion (*Figure 2D,E*). Volcano plots show that almost all families underwent significant up-regulation from D0 to D6 (*Figure 2D*, left panel), ranging from modest (LINEs) to robust (MMERGLN) fold changes. Silencing restoration also occurred globally between D6 and D13, except for IAPEy or B1 elements, which remained at constant levels (*Figure 2D*, middle panel). Comparison of transposon expression levels between the two end-points (D0 and D13) indicated skewing in both directions (*Figure 2D*, right panel). Some families, such as MERVL, SINEs B2 or any LINE1 types, were more strongly repressed after the 2i+vitC conversion at D13 than initially at D0 in serum. Others, like MMERGLN, ETnERV3 and IAPEz, underwent repression from D6 to D13, but not to the full extent when compared to D0. As a general rule, these data show that non-LTR (LINEs and SINEs) and LTR elements belonging to the K, L and 1 classes—albeit very different in terms of evolutionary origins and genomic structures—adopt common fates upon acute loss of DNA methylation.

To examine whether the burst of transcription observed from bulk RNA profiling emanated from a few discrete elements or reflected a general trend within each family, we measured the transcriptional output of individual transposon copies by allowing unique read mapping only. We found that 7163 uniquely identifiable LINEs and 2372 ERVs showed activity throughout the conversion, which represented 1.2% and 3.8% of the total number of LINE1s and ERVs, respectively, or 19.3% and 15.2% of the intact elements of these families (integrity score >0.8). Importantly, these numbers are likely underestimated because active but identical copies cannot be discriminated, and are discarded from the analysis. Within all families, the number of significantly expressed elements was higher at D6 than at D0 or D13 of conversion (*Supplementary file 2A* and *Figure 2—figure supplement 4C*). Generally, not only were more elements active, but individual copies also gained expression at D6 (*Figure 2F* and *Figure 2—figure supplement 4D,E*). Finally, active transposons were evenly distributed along chromosomes, with no particular genomic hotspot (*Figure 2—figure supplement 4F*).

As a whole, the unique mapping analysis confirmed the class-specific features previously inferred from the familial analysis, regarding the degree of activation at D6 (from modest for LINEs to intense for MMERGLN) and silencing restoration at D13 (strong for LINEs, intermediate for MMERGLN and nonexistent for IAPEy). Most importantly, it uncovered unprecedented details into the diverse regulation of individual transposons. Expression levels were the most homogeneous among elements of the same family during the D6 de-repression phase. Comparatively, at the D13 silencing restoration time-point, we observed heterogenic regulation at the inter- and intra-familial levels (*Figure 2F* and *Figure 2—figure supplement 4D*). Some families, such as LINEs and MMERGLN, displayed

collective behaviors, with the vast majority of elements simultaneously undergoing repression. In contrast, IAPEz, MERVL or ETn elements showed the widest distribution in individual expression. In particular, IAPEz elements were split into two categories at D13, one that maintained high expression, and the other that underwent complete silencing.

Globally, our analysis reveals that transposons undergo a transient relaxation of silencing upon DNA methylation loss followed by an expression reduction phase. However, family- and element-specific behaviors provide nuance to this general trend. It should be stressed here that a certain degree of heterogeneity is frequently inaccessible for young and highly conserved families of transposons, such as IAPEz and MMERVK10C, for which mapping reads to precise genomic locations is ambiguous, if not impossible.

## Silencing release is specific to transposons

Compared to transposons, protein-coding genes followed different dynamics during 2i+vitC conversion (*Figure 3A* and *Figure 3—figure supplement 1A*). The vast majority exhibited stable expression, while 3,301 genes were either up- or down-regulated; these numbers are similar to previous reports of a serum to 2i transcriptional switch (*Marks et al., 2012*). While the general expression trend for transposons was biphasic, most differentially expressed genes displayed a monotonic pattern. A relevant example is the *Dazl* gene, which was continuously up-regulated from D0 to D13 (*Figure 3B*), reflecting its sensitivity to vitC (*Blaschke et al., 2013*). Conversely, expression of genes encoding transcription factors of the ZSCAN4 family progressively decreased during the conversion, with undetectable transcripts by D13 (*Figure 3B*). As expression of these factors reflects a subpopulation of ES cells exhibiting a transcriptional profile akin to 2-cell stage embryos (*Macfarlan et al., 2012*), our results imply that 2-cell-like cells exist in serum-based conditions but disappear in 2i+vitC medium.

The burst of expression at D6 appears specific to transposons, and is not a general trend of the genome. Nevertheless, 156 genes adopted a transposon-type pattern, with a peak at D6 followed by subsequent down-regulation at D13 (*Figure 3A* and *Supplementary file 1*). These genes were linked to ontology categories such as organismal development and were significantly enriched for transcription factors, most notably those related to pluripotency (*Supplementary file 2B*). Further examination indicated that transcription of *Nanog*, *Klf4*, *Tbx3* and *Prmd14* peaked at D6 (*Figure 3C*); enhanced production of pluripotency-related proteins was also observed by western blot within the first few days of 2i+vitC conversion (*Figure 3D*). Therefore, the peak of transposon transcription at D6 coincides with a maximum availability of pluripotency regulators.

It was previously shown that LTR sequences of ERVs can direct transcription of nearby genes in ES cells and early embryos, forming chimeric transcripts (*Karimi et al., 2011b*; *Macfarlan et al., 2012*). We detected several dozens of genes that used a promoter located in a transposable element (ERV or LINE1), independently of the medium composition (*Supplementary file 2C*). A particular case was *Mep1b*, which clusters with the 156 'transposon-like' genes. This gene was induced tenfold at D6, concomitant with the activation of the RLTR9E element driving its expression, before returning to its initial level at D13 (*Figure 3E*). Several 2-cell-specific genes have been shown to initiate from class L LTRs (*Macfarlan et al., 2011*); some of them, like *Ubftl1*, were indeed specifically overexpressed at D6; in contrast, *Zscan4d* expression was uncoupled from the expression of the adjacent full-length MERVL element in our system (*Figure 3—figure supplement 1B,C*). Although the burst of transposon expression at D6 can sporadically coordinate the transient activation of adjacent genes, it can be concluded that the genome-wide relaxation of transposons had generally a minimal effect on protein-coding gene expression.

## Reconfiguration of the repressive chromatin landscape upon 2i+vitC conversion

To gain insight into the basis for transposon regulatory dynamics, we examined the chromatin state of cells undergoing serum to 2i+vitC conversion. By western blot and immunostaining, we observed large-scale reorganization of histone modifications linked to transcriptional repression. While H3K9me3 marks remained globally constant, H3K9me2 levels were strongly reduced and, inversely, H3K27me3 levels increased from the first days of conversion (*Figure 4A* and *Figure 4—figure supplement 1A*). The global dynamics of histone marks were not correlated with the changes in the

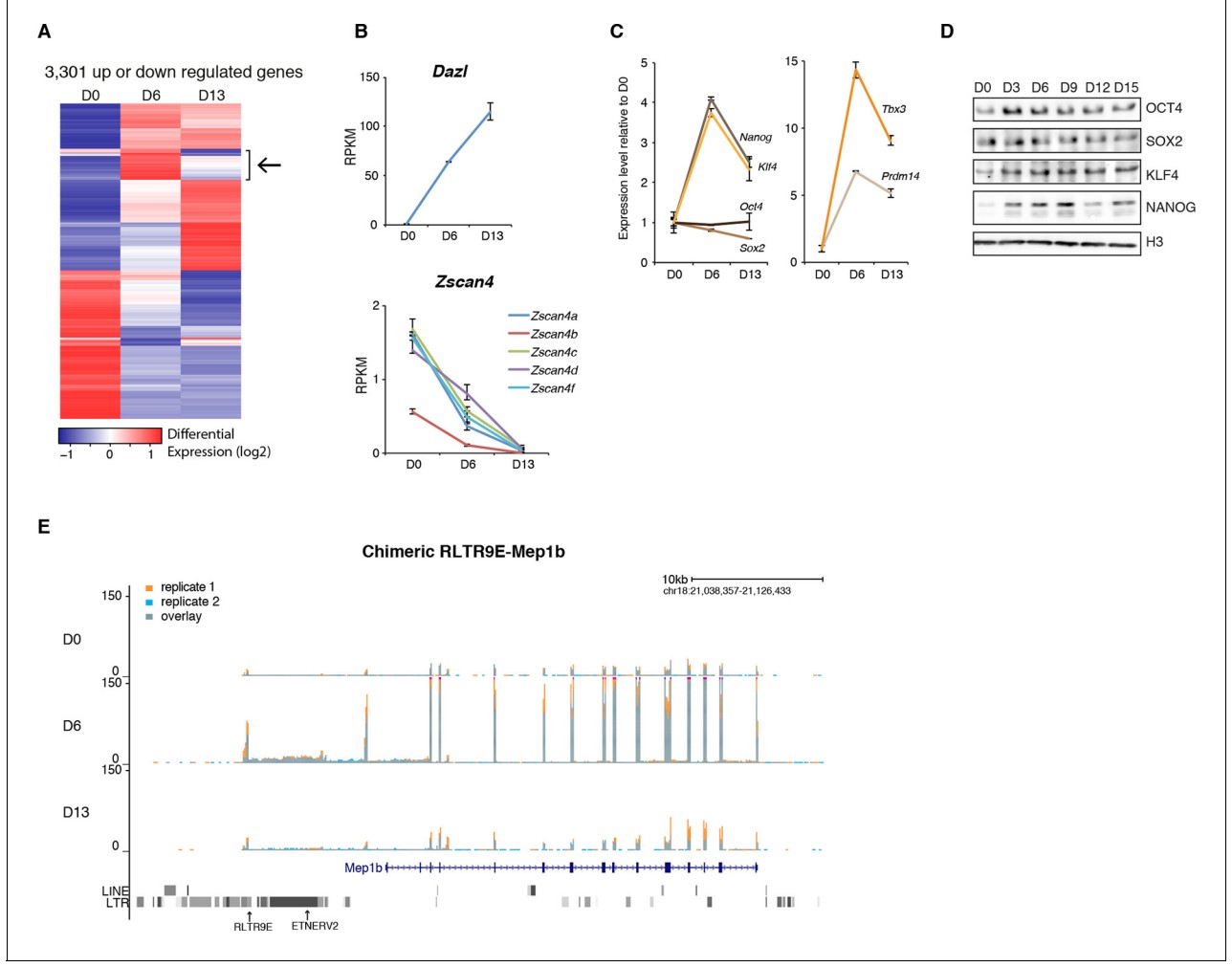

**Figure 3.** Gene expression analysis upon serum to 2i+vitC conversion. (**A**) Heatmap representation of genes (*n* = 3301) that are significantly misregulated between at least two time points of D0, D6 and D13 during medium conversion. Color codes as in *Figure 2E*. The arrow highlights a subset of genes whose expression transiently peaks at D6. (**B**) Monotonic expression patterns of *Dazl* and *Zscan4* family genes as measured by RNA-seq at D0, D6 and D13, expressed in RPKM (read per kb per millions). Mean ± SEM between two biological replicates. (**C**) Dynamic expression of pluripotency transcription factor genes as measured by RNA-seq. Data is expressed as fold change to D0 and represent mean ± SEM between two biological replicates. (**D**) Expression of core pluripotency transcription factors by western blot. (**E**) RNA-seq track showing a chimeric transcript between a RLTR9E transposon element and the *Mep1b* gene specifically expressed at D6. Data represent normalized read density.

The following figure supplement is available for figure 3:

**Figure supplement 1.** Gene expression analysis upon serum to 2i+vitC conversion.

availability—or lack thereof—of H3K9me2 modifiers and components of the polycomb machinery (*Figure 4—figure supplement 1B*).

ChIP-qPCR measurement confirmed either the constitutive absence or the rapid removal of H3K9me2 at several transposon types upon 2i+vitC conversion (*Figure 4B*), making it unlikely that this mark could participate to long-term transposon silencing in the absence of DNA methylation. To functionally exclude this possibility, we examined ES cells lacking the EHMT2 H3K9 dimethyltransferase (*Tachibana et al., 2002* and *Figure 4—figure supplement 1C*). As previously described, when cultured in serum, *Ehmt2*-KO ES cells did not exhibit significant up-regulation of transposons as measured by Nanostring, with the exception of MERVL elements (*Macfarlan et al., 2012*; *Maksakova et al., 2013*). Upon 2i+vitC conversion, while LINE1 elements behaved as in WT cells, IAPEz and MERVL expression was enhanced around D6 in *Ehmt2*-KO cells (*Figure 4—figure*

supplement 1C). As H3K9me2 cannot be detected after D3 at these sequences (*Figure 4B*), this relative up-regulation likely occurs through indirect effects. Most importantly, *Ehmt2* mutants exhibited transposon re-silencing after D6, which excludes a role for H3K9me2 in compensating the loss of DNA methylation-dependent repression.

We then focused our analysis on the distribution of H3K9me3 and H3K27me3 marks by chromatin immunoprecipitation followed by deep sequencing (ChIP-seq) in biological replicates at D0, D6 and D15 of conversion, allowing multiple mapping with random allocation (*Supplementary file 1* and *Figure 4—figure supplement 2A*). Neither the total number of H3K9me3 peaks (39424 at D0 and 38554 at D15), nor their preferential occurrence on transposons was significantly altered during the conversion (*Figure 4C*). In contrast, the number of H3K27me3-enriched regions raised four fold from D0 to D15 (9663 to 40098). The vast majority of newly gained H3K27me3 peaks were located in ERV and LINE1 repeats, at the expense of gene promoters (*Figure 4C*). We also observed a gradual H3K27me3 re-localization to pericentric heterochromatin during the conversion, by ChIP-qPCR at major satellite repeats (*Figure 4D*), by immunostaining (*Figure 4—figure supplement 1A*) and by mapping ChIP-seq reads to the major satellite consensus sequence (*Figure 4—figure supplement 2B*). Redistribution of H3K27me3 from gene promoters towards satellite repeats was previously reported in 2i-only conditions (*Marks et al., 2012*). However, increased H3K27me3 levels and subsequent accumulation at different transposon repeats seems specific to the globally hypomethylated genome of 2i+vitC-cultured cells. Accordingly, hypomethylated *Dnmt*-tKO ES cells grown in serum displayed similar H3K27me3 redistribution towards transposons when we analyzed available ChIP-seq data (*Figure 4—figure supplement 2C*).

In agreement with the high expression of these elements, the 5' UTR of LINEs (L1-A and L1-T) and the LTR of IAPEz were enriched in H3K4me3, as assessed by ChIP-qPCR (*Figure 4—figure supplement 2D*). Interestingly, bisulfite-pyrosequencing analysis of H3K4me3- or H3K9me3-immunoprecipitated chromatin confirmed, as expected, that active LINE1 elements (marked by H3K4me3) had lower DNA methylation levels compared to inactive ones (marked by H3K9me3)(*Figure 4—figure supplement 2E*). This provides an additional documentation of the intra-familial heterogeneity of LINE1 elements at D6.

## Relative H3K9me3 and H3K27me3 enrichments define three categories of transposons

We next measured relative H3K9me3 and H3K27me3 levels over different transposon families, focusing our analysis on elements that were scored as intact. At D0 in serum, most transposon families were occupied by H3K9me3 to various extents, but lacked H3K27me3 (*Figure 4E*). One noticeable exception was RLTR4, which exhibited a strong H3K27me3 signal (*Figure 4—figure supplement 3A*). Interestingly, this element is 90% identical to MuLV, which is one of the few transposons up-regulated in polycomb-deficient ES cells (*Leeb et al., 2010*). Upon 2i+vitC conversion, H3K9me3 levels remained largely constant, although patterns observed in serum tended to be exacerbated: families with the initial highest enrichment (IAPEz, RLTR6 and MMERVK10C) were further enriched for this mark, while families with modest enrichment (MERVL, MURVY or the MalR-class L ORR1A and ORR1B) tended to lose it. Meanwhile, H3K27me3 progressively accumulated at most transposons (*Figure 4E*), and this gain was variable among families: from inexistent for IAPEz to moderate for LINEs and VL30, and to strong for various ERVs.

Of note, when focusing on ERV elements, high levels of H3K9 or H3K27 methylation were found to be restricted to full-length intact elements: isolated solitary (solo) LTRs (not interspersed with other repeats) had most often no enrichment for H3K9 or H3K27 methylation (*Figure 4—figure supplement 3C*). This suggests that internal ERV sequences are important for H3K9me3 and H3K27me3 recruitment. IAP elements were an exception, as their solo-LTRs were enriched for H3K9me3 on their own.

Remarkably, different kinetics were observed for H3K9me3- and H3K27me3-related changes: H3K9me3 levels were rapidly modified between D0 and D6, while H3K27m3 gain lagged behind, reaching its full extent between D6 and D15. Although the whole picture is quite complex, it can be concluded that medium-induced DNA methylation profoundly remodels the repressive chromatin landscape of transposons. From a universal H3K9me3 occupancy in serum, transposon families exhibited three general trends in 2i+vitC: A) co-occupancy of H3K9me3 and H3K27me3 (LINEs, MMERGLN, RLTR6, RLTR10, IAPEy, VL30), B) exclusive H3K9me3 occupation (IAPEz, MMERVK10C),

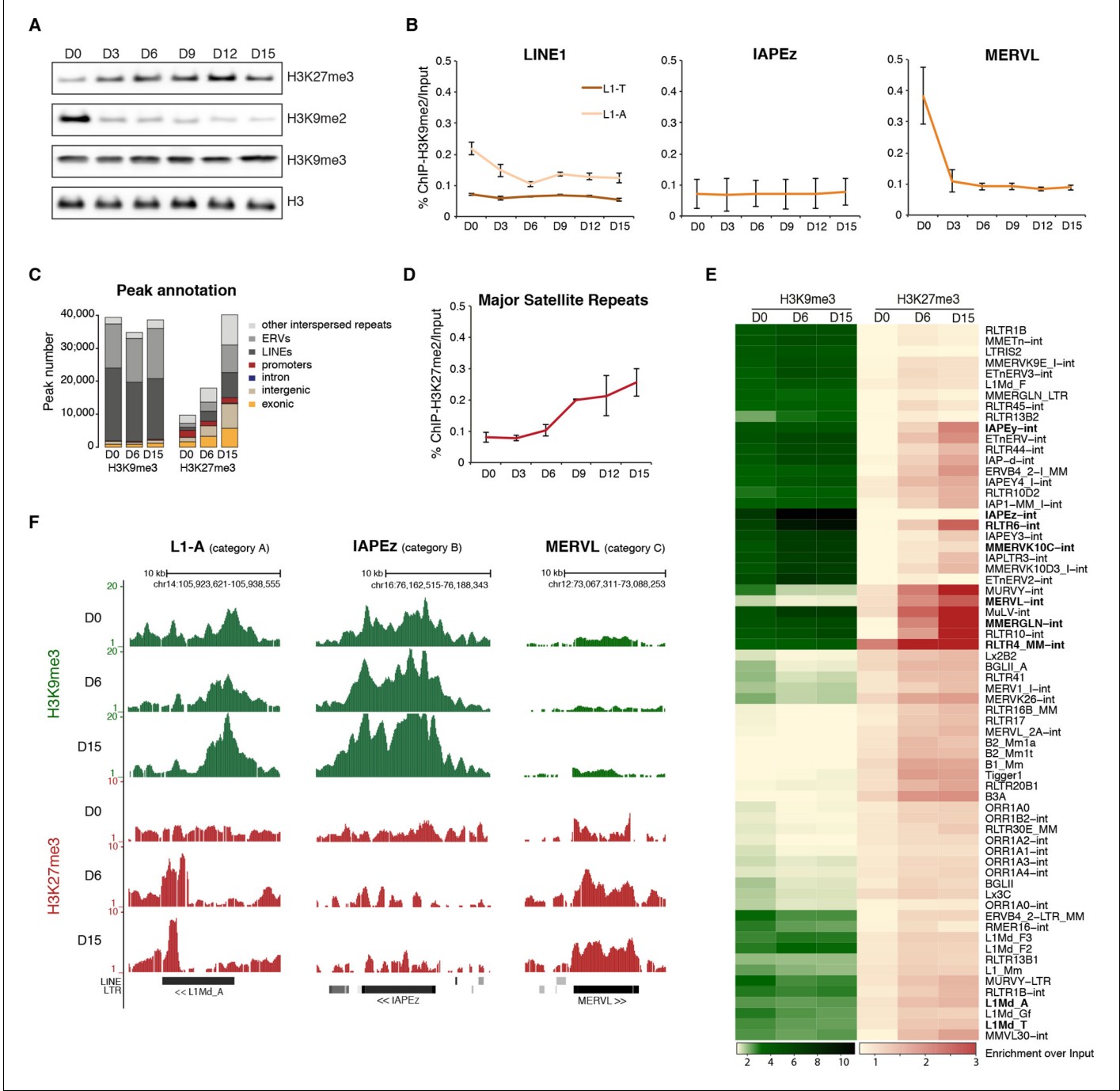

**Figure 4.** Repressive chromatin reorganization upon loss of DNA methylation. (**A**) Western blot analysis of global levels of repressive histone modifications during the course of serum to 2i+vitC conversion. (**B**) H3K9me2 enrichment levels at three transposon families as measured by ChIP-qPCR. Quantitative data are expressed as the percentage of ChIP over Input. Data represents mean ± SEM of two biological replicates. (**C**) Genomic annotation of ChIP-seq peaks. Data represent the number of annotated peaks. H3K9me3 data are representative of two biological replicates, while H3K27me3 data represent only one, as peak calling could not be performed successfully on the second replicate. (**D**) H3K27me3 enrichment at major satellite repeats as measured by ChIP-qPCR. Data are represented as in 4B. (**E**) Heatmap and hierarchical clustering of average H3K9me3 and H3K27me3 levels in 69 transposons families at D0, D6 and D15. Colors represent the average read count for an element in a given family, relative to input (average between two biological replicates). Only intact (score>0.8) elements were considered. (**F**) Representative genomic region depicting evolution of H3K9me3 (green) and H3K27me3 (red) at L1-A (category A), IAPEz (category B) and MERVL (category C) transposons. Data represent normalized read density.

*Figure 4 continued on next page*

Figure 4 continued

The following figure supplements are available for figure 4:

**Figure supplement 1.** Repressive chromatin reorganization during conversion from serum to 2i+vitC.

**Figure supplement 2.** Repressive chromatin reorganization during conversion from serum to 2i+vitC.

**Figure supplement 3.** Screenshots of repressive chromatin reorganization at transposons.

and *C)* complete switch from H3K9me2/3 to H3K27me3-regulated chromatin (MERVL and MURVY) (*Figure 4E,F* and *Figure 4—figure supplement 3B*). Our analysis therefore provides a classification of the different transposon families into three main categories (A, B, and C), according to the chromatin signature they adopt upon DNA methylation loss.

To assess the behavior of individual elements among these three generic patterns, we attempted to analyze unique mappers, but the coverage on individual transposons was too low to extract reliable information. Nevertheless, to gain insight into the question of intra-familial heterogeneity, we plotted H3K9me3 and H3K27me3 enrichment for every intact transposons (score>0.8) per family during the conversion. We found that elements of the B and C categories tended to be very homogeneous. IAPEz elements (category B) collectively gained H3K9me3 from D0 to D6; the MERVL and the Y-specific MURVY families (category C) also showed compact patterns, with individual elements transitioning together from H3K9me3 enrichment at D0 to H3K27me3 at D15 (*Figure 5A* and *Figure 5—figure supplement 1C*). The A category, which is enriched for both H3K9me3 and H3K27me3, was more diverse, with some families displaying homogeneous patterns, while others showed intra-familial dispersion in chromatin fates upon conversion. Within the MMERGLN, RLTR6 or RLTR10 families, all elements gained H3K27me3 while maintaining or gaining high levels of H3K9me3. Within the L1-T and IAPEy families, the majority of elements gained H3K27me3, but a subset maintained H3K9me3 without acquiring H3K27me3 (*Figure 5A* and *Figure 5—figure supplement 1A*). Another case of intra-familial heterogeneity is provided by RLTR4, which specifically carries H3K27me3 marks at D0 in serum: we demonstrate here that this enrichment was restricted to a small proportion of elements, as was previously suspected (*Reichmann et al., 2012* and *Figure 5—figure supplement 1D*). By extracting single-element information from RNA-seq and ChIP-seq data, it is clear that both transcriptional and chromatin heterogeneity exists among some transposon families. Our analysis reveals that caution should be taken when interpreting average familial behaviors, as they may be representative of only a few individual elements inside a given family.

## H3K27me3 occupies DNA methylation- and H3K9me3-free territories of transposon sequences

H3K27me3 and H3K9me3 marks usually do not occur concomitantly (*Mikkelsen et al., 2007*). We were therefore intrigued to observe that H3K27me3 and H3K9me3 were simultaneously enriched at transposon families of the A category in 2i+vitC medium (*Figure 4E* and *Figure 5A*). To map the relative position of H3K9me3 and H3K27me3, we determined their average profile over full-length individual elements of all transposon families, including their immediate genomic vicinity (+/- 5 kb from the center of each element). Notably, H3K9me3 domains often spread out on adjacent genomic regions, whereas H3K27me3 was confined to transposon sequences (*Figure 5B,C* and *Figure 5—figure supplement 2A*). It was previously described that H3K9me3 enrichment is restricted to the 5' UTR of LINEs, while being evenly distributed along the entire length of ERVs (*Bulut-Karslioglu et al., 2014*; *Pezic et al., 2014*). In fact, we found this to be valid for specific ERVK elements only, namely IAPEz, IAPEy and MMERKV10C.

Our most striking finding was the observation of a spatial separation between the two marks in category A transposons: H3K9me3 tended to occupy the 5' end, while H3K27me3 preferentially targeted the 3' end. This was observed for a significant proportion of LINEs and for several ERVs of the 1 or K classes (MMERGLN, RLTR6, MuRRS, RLTR10) (*Figure 5C* for visual examples). However, some category A families showed H3K9me3 and H3K27me3 co-localization in their 5' region (VL30, IAPEy, ETnERV). Notably, these transposon families harbor the greatest individual chromatin heterogeneity

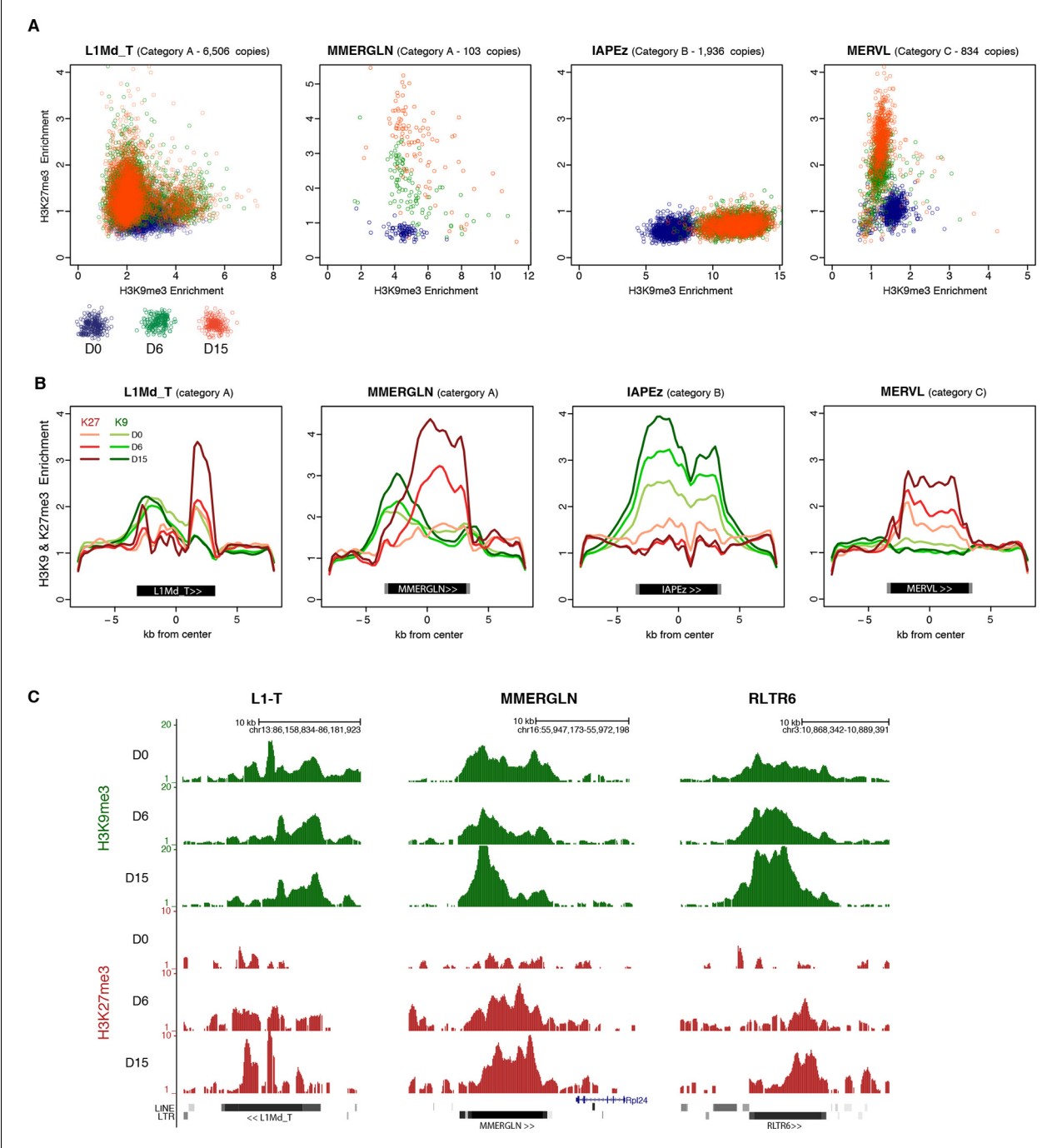

**Figure 5.** H3K9me3 and H3K27me3 mark the same transposons but do not spatially overlap. (**A**) Normalized H3K9me3 and H3K27me3 enrichment over input at individual elements from different transposon families. Each dot represents a single element at D0 (blue), D6 (green) and D15 (red). Only intact (integrity score>0.8) elements were considered. Data represent the average between two biological replicates. Analyzed numbers of distinct transposon copies per family appear into brackets. (**B**) Composite profile showing H3K9me3 (green) and H3K27me3 (red) coverage along different transposon sequences at D0, D6 and D15 of medium conversion. (**C**) Representative genomic regions comprising LINE1 and ERV repeats that gain H3K27me3 in their 3' end during the conversion, while maintaining H3K9me3 in the 5' end. Data represent normalized read density.

The following figure supplements are available for figure 5:

**Figure supplement 1.** Intra-familial heterogeneity of H3K9me3 and H3K27me3 in transposons.

**Figure supplement 2.** H3K9me3 and H3K27me3 mark transposon sequences but do not spatially overlap.

upon conversion (*Figure 5—figure supplement 1A*): we presume that the metaplot figures likely represent an average among different individual elements and/or different cell populations.

Having demonstrated that culture-induced DNA demethylation leads to increased and family-specific distribution of H3K27me3 on transposon sequences, we reasoned that similar features might occur upon genetically induced DNA demethylation. Fulfilling this prediction, analysis of available ChIP-seq datasets (*Brinkman et al., 2012*) showed concordant H3K27me3 patterns between serum-grown *Dnmt*-tKO ES cells and 2i+vitC-grown ES cells: entire coverage of MERVL sequences, 3' localization in MMERGLN and RLTR6, and 5' localization in VL30 and ETnERV (*Figure 5—figure supplement 2B*). These results support the notion that the pattern of H3K27me3 distribution on transposon sequences corresponds to an adaptation to the lack of DNA methylation.

In summary, upon 2i+vitC-induced DNA demethylation, H3K27me3 and H3K9me3 can converge on category A transposon sequences, but they occupy different territories. IAPEz (category B) and MERVL (C) represent extreme cases of exclusivity, with the former being entirely covered by H3K9me3, and the latter by H3K27me3. Our study provides unprecedented evidence that H3K27me3 deposition at transposons is a default response to the absence of both DNA and H3K9 methylation.

## Chromatin silencing pathways play diverse roles at transposons

Our analysis reveals that H3K9me3 and H3K27me3 jointly or separately decorate transposon sequences upon DNA methylation loss. Through genetic analyses, we aimed to discern the functional relevance of these marks in controlling the three categories of transposons that we defined. Regarding H3K9me3-dependent pathways, we used CRISPR/Cas9 editing to generate a double-knockout ES cell line for the H3K9 trimethyltransferases, SUV39H1 and SUV39H2 (*Suv39h*-dKO). Additionally we created haploinsufficient mutants for H3K9 trimethyltransferase SETDB1 (*Setdb1* +/-) and its TRIM28 co-repressor (*Trim28* +/-) (*Figure 6—figure supplement 1A,B*); complete SETDB1 or TRIM28 removal is not compatible with ES cell survival (*Dodge et al., 2004*; *Rowe et al., 2010*). The role of H3K27me3 was studied in mutant ES cells for the EED protein (*Eed-KO, Schoeftner et al., 2006*), which is required for H3K27me3 catalysis by the Polycomb Repressive Complex 2 (PRC2) (*Margueron and Reinberg, 2011*). *Eed*-KO ES cells experienced massive cell death around D8 of conversion, but slowly recovered in the following days (data not shown). Finally, in order to study compensatory mechanisms between H3K9 and H3K27 pathways, we deleted EED in *Suv39h*-dKO ES cells, generating *Suv39-Eed* triple-knockout lines (*Suv39h-Eed*-tKO) (*Figure 6—figure supplement 1C*). *Suv39-Eed*-tKO ES cells were viable with a reduced proliferation rate when grown in serum, but did not survive more than a week when converted to 2i+vitC (data not shown). Nanostring quantification of transposon transcripts was performed upon serum to 2i+vitC transfer of these five cell lines, which, importantly, share the same J1 cell background.

As representatives of transposon A category, young LINE1 elements maintain H3K9me3 while gaining H3K27me3 during medium conversion. Previous studies concluded that L1 repression in serum relies on SUV39H-dependent H3K9me3 (*Bulut-Karslioglu et al., 2014*): through the analysis of our genetic mutants, we found this was the case for L1-A, and very modestly for L1-T elements (*Figure 6A* and *Figure 6—figure supplement 2A*). Moreover, LINE1 expression was significantly upregulated at the end of the conversion of *Setdb1* +/- ES cells, suggesting an important role for SETDB1-mediated repression upon DNA methylation loss. In regards to the role of polycomb, RNA-seq analysis of *Eed*-KO ES cells revealed that absence of polycomb had little effect on transposons in ES cells with a methylated genome (at D0 of conversion). In contrast, many transposon families, including the different young LINE1 subtypes that belong to the A category, were strongly activated in fully 2i+vitC-converted *Eed*-KO cells (*Figure 7A,B* and *Figure 7—figure supplement 1*): polycomb-mediated silencing is therefore involved in controlling these transposons, specifically in absence of DNA methylation. Interestingly, ERV families that gain H3K27me3 uniquely in their 3' end – such as MMERGLN, RLTR6 or RLTR10 – were not misregulated in absence of EED; this suggests that 3' end deposition of H3K27me3 does not drive their transcriptional repression.

The category B of transposons is exemplified by IAPEz elements, which harbor exclusive H3K9me3 enrichment in all culture conditions. Although this profile would predict a continuous and exclusive dependence towards H3K9me3 upon medium adaptation, we observed complex behaviors in the different mutants (*Figure 6B*). During conversion, *Trim28+/-* and *Setdb1+/-* cells showed enhanced IAPEz up-regulation and repression failure after D6, in line with a major role of SETDB1-

related H3K9me3 for controlling these elements in ES cells (*Matsui et al., 2010*). However, SUV39H depletion led to an unexpected IAPEz suppression upon conversion. One possible explanation is that *Suv39h*-dKO cells have acquired long-term compensatory mechanisms that prevent transient IAPEz activation upon DNA methylation loss. Moreover, IAPEz elements were more strongly expressed in *Eed*-KO compared to WT cells during conversion (*Figures 6B* and *Figure 7A,B*), which is at odds with their apparent lack of H3K27me3 enrichment in ChIP-seq data (*Figures 4E,F*). Analysis of individual elements showed that this activation did not emanate from a few discrete elements but represented a general trend (*Figures 7C*). These results could be due to indirect effects of the *Eed* deficiency.

Finally, the H3K9me2/3- to H3K27me3-chromatin transition undergone by category C elements was very clearly illustrated in chromatin modifier mutants (*Figure 6C*). MERVL elements are known to be primarily repressed by EHMT1/EHMT2-dependent H3K9me2 marks in serum (*Maksakova et al., 2013*). We found that MERVL silencing also strongly relied on SUV39H-control in serum (*Figure 6C*), which correlates with a modest H3K9me3 enrichment in our ChIP-seq data. SUV39H-dependent H3K9me3 became dispensable for MERVL silencing upon 2i+vitC conversion, and the switch towards H3K27me3 control was perfectly correlated with a 15-fold expression increase in *Eed*-KO cells (*Figure 6C*). RNA-seq analysis indicates that MERVL was actually the most highly activated transposon in absence of EED, as a result of a collective upregulation of all individual elements (*Figure 7A–C*). MERVL therefore represents a striking model of epigenetic switch from H3K9 to H3K27 methylation-based repression, which occurs subsequently to DNA methylation loss. Interestingly, we found that polycomb-dependent control can be implemented in absence of SUV39H, even in cells with high DNA methylation levels: while MERVL elements were upregulated by five-fold in *Suv39h*-dKO cells in serum grown-conditions, additional depletion of EED in *Suv39-Eed*-tKO cells led to a 20 fold increase compared to WT cells (*Figure 6C*).

Finally, MERVL silencing was previously shown to rely on other histone modifiers, such as histone deacetylases (HDACs) (*Macfarlan et al., 2011*). Treatment with the HDAC inhibitor Trichostatin A (TSA) resulted in strong upregulation of MERVL in both serum and 2i+vitC conditions (*Figure 6—figure supplement 2B*), suggesting that histone deacetylation is required for efficient silencing of MERVL, in presence or absence of DNA methylation.

## Discussion

Our study provides unprecedented insight into the dynamic adaptation of the pluripotent genome to a loss of DNA methylation-based control of transposons. This was achieved through detailed kinetic assessment of transcription and chromatin states during conversion of WT ES cells from serum to 2i+vitC media, as a way to reproduce the DNA methylation erasure that occurs during embryogenesis. Despite their heterogeneous origins and structures, we found that various transposon families residing in the mouse genome adopted a common regulatory fate: after an initial transcriptional burst, repression was re-established in a DNA methylation-independent manner. Distinct combinations of H3K9me2/3 and H3K27me3 were observed among transposon families, defining three functional categories of chromatin-based responses to DNA methylation loss: joint H3K9me3 and H3K27m3 (A), H3K9me3-exclusive (B), and H3K27me3-exclusive (C) (*Figure 8*). Importantly, *Dnmt*-tKO cells, which have endured long-term adaptation to a DNA methylation-free state, displayed similar transposon-specific chromatin patterns when grown in serum, which excludes a medium-related effect. In conclusion, our work revises the previous assumption that DNA methylation is dispensable for transposon silencing in ES cells; rather, we reveal here that various histone-based repression strategies are implemented upon DNA methylation loss, thereby safeguarding pluripotent cells against a multitude of heterogeneous transposon entities.

Upon 2i+vitC-mediated DNA methylation loss, the repertoire of repressive histone marks is profoundly remodeled (*Figure 8A*): H3K9me2 enrichment decreases, while H3K27me3 is enhanced and H3K9me3 levels are globally constant. Interestingly, repressive chromatin reorganization has also been cytologically observed in primordial germ cells (PGCs) undergoing genome-wide demethylation (*Hajkova et al., 2008*; *Seki et al., 2007*). Moreover, the relocalization of H3K27me3 at transposons and its co-occurrence with H3K9me3 was also reported in hypomethylated PGCs (*Liu et al., 2014*). Our cellular system could therefore represent an adequate model to study *in vivo* events of chromatin reprogramming occurring at transposons upon DNA methylation loss.

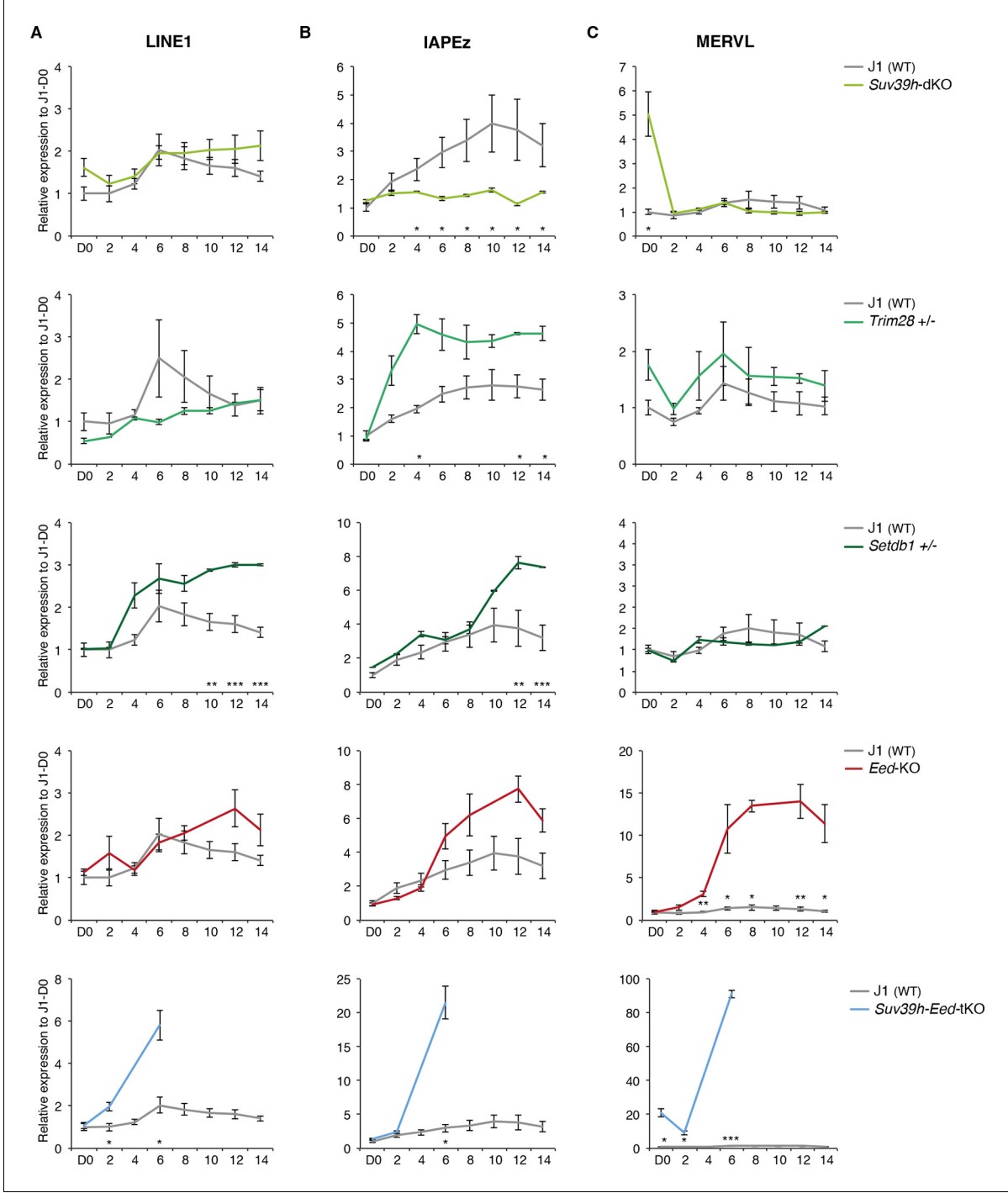

**Figure 6.** Complex regulation of transposons by SUV39H, TRIM28, SETDB1 and EED upon loss of DNA methylation. Expression levels in *Suv39h*-dKO, *Trim28+/-*, *Setdb1+/-*, *Eed*-KO, *Suv39h-Eed*-tKO and WT J1 ES cells for: (**A**) LINE1 (category A) (**B**) IAPEz (category B) (**C**) MERVL (category C) Expression levels were measured by Nanostring nCounter. Data are expressed as fold changes to WT D0 and represent mean ± SEM between two (*Trim28* and *Setdb1*), three (*Suv39h* and *Suv39h-Eed*-tKO), four (*Eed*-KO) and eight (J1) biological replicates. *p<0.05, **p<0.01 and ***p<0.001 (unequal variances t-test between WT and mutants at a given day).

The following figure supplements are available for figure 6:

**Figure supplement 1.** Caracterization of genetic mutants.

**Figure supplement 2.** Regulation of transposons by HDACs

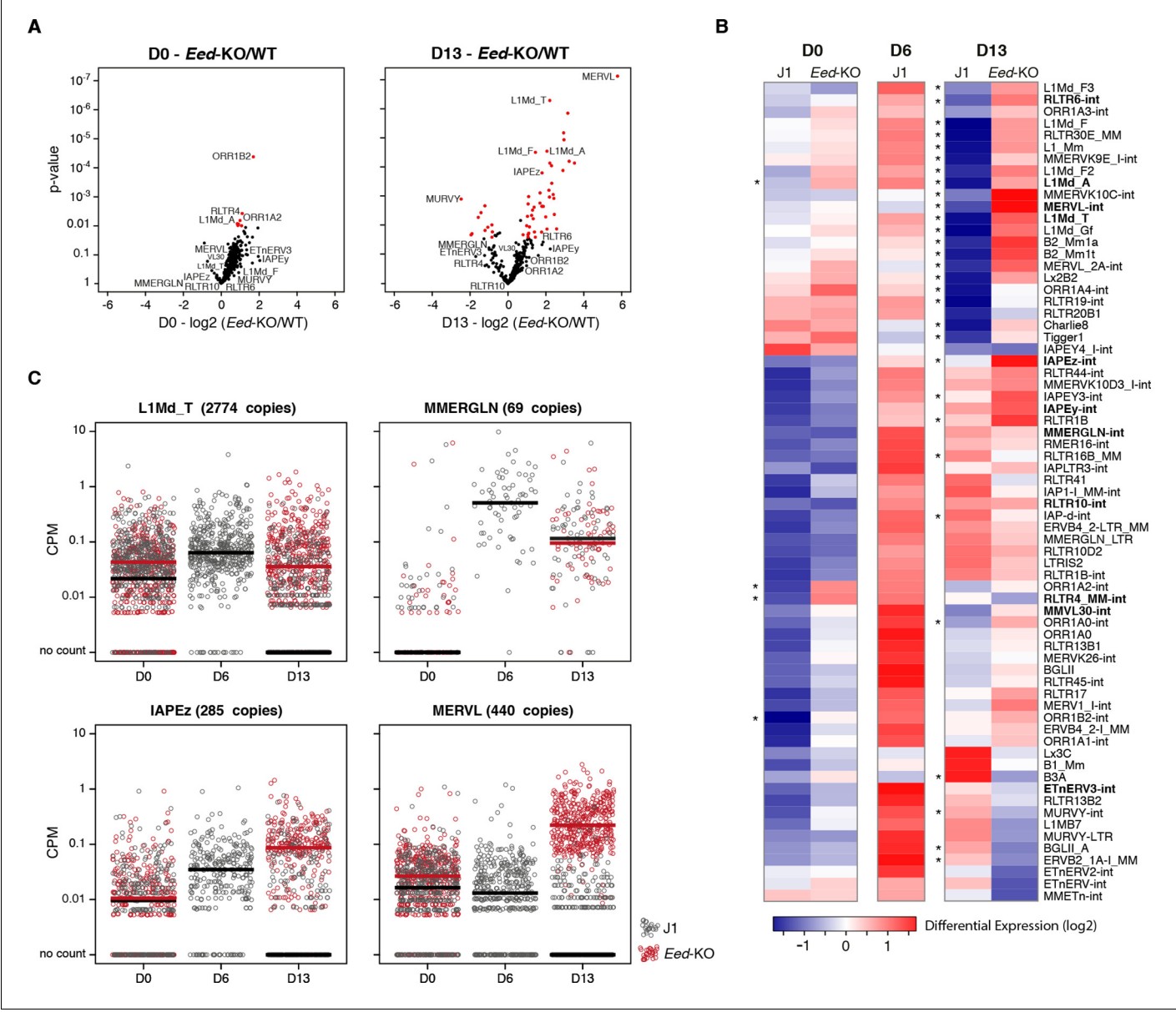

**Figure 7.** Polycomb regulates transposons in absence of DNA methylation. (**A**) Volcano plot representation of up- and down-regulated transposons as measured by RNA-seq between WT and *Eed*-KO cells at D0 (left) and D13 of conversion (right). Red dots indicate significantly misregulated repeats between two conditions (fold change >2 and p-value <0.05). (**B**) Heatmap representation and hierarchical clustering of expression changes for 69 transposon families at D0, D6 and D13, in WT and *Eed*-KO cells. Colors represent on a log2-scale the differential expression between a given condition and the average of the five conditions. *p<0.05. (**C**) Expression of individual elements from different transposon families at D0, D6 and D13, in WT and *Eed*-KO cells, expressed in CPM. The black and red bars represent the median of the distribution, for WT and *Eed*-KO, respectively.

The following figure supplement is available for figure 7:

**Figure supplement 1.** RNA-seq of *Eed*-KO ES cells

The persistence of H3K9me3 upon loss of DNA methylation highlights that DNA methylation does not exert significant control over H3K9me3-targeting of transposons in ES cells. Interestingly, we consistently observed that regions of persistent DNA methylation (RMRs) coincide with high H3K9me3 enrichment on transposon sequences in fully 2i+vitC-converted cells, *e.g.* on the 5' end of LINE1 elements and throughout the length of ERVK elements. This supports previous evidence that H3K9me3 can confer protection against DNA demethylation (*Leung et al., 2014*). Inversely, the

rapid disappearance of H3K9me2 upon serum to 2i+vitC conversion could reflect a direct role of DNA methylation in the maintenance of these marks. Accordingly, H3K9me2 reduction was also observed in DNA methylation-free *Dnmt*-tKO ES cells grown in serum (data not shown). Coupled losses of DNA methylation and H3K9me2 have also been previously reported *in vivo*, during normal primordial germ cell development (*Hajkova et al., 2008*; *Seki et al., 2005*) and in DNA methylation-deficient spermatocytes (*Zamudio et al., 2015*). A mechanism of H3K9me2 methyltransferase recruitment via DNA methylation has been resolved in plants (*Du et al., 2015*); the evolution of an analogous mechanism in mammals should be explored.

Of particular importance to this study is our observation of an epigenetic switch occurring between DNA methylation- and H3K27me3-based control. H3K27me3 is barely detectable at transposons in DNA hypermethylated WT ES cells grown in serum; in contrast, transposons accumulate H3K27me3 in both 2i+vitC-converted cells and in serum-grown *Dnmt*-tKO cells. This is in line with the prevailing notion that DNA methylation and H3K27me3 are mutually exclusive genome-wide and that DNA methylation antagonizes H3K27me3 deposition (*Brinkman et al., 2012*; *Jermann et al., 2014*; *Tanay et al., 2007*). Saliently, this raises the question as to how transposons acquire H3K27me3 upon DNA methylation loss. In mammalian genomes, polycomb is typically targeted to unmethylated GC-rich gene promoters (*Jermann et al., 2014*; *Mendenhall et al., 2010*). Notably, transposon sequences have a GC content superior to the genome average (*Figure 5—figure supplement 2B*): this signature could be sufficient to attract polycomb-mediated H3K27me3 deposition in the absence of DNA methylation. Intermediate methyl-sensitive DNA binding proteins may be involved: the BEND3 protein was recently identified as a sensor of DNA methylation states at pericentromeric repeats, recruiting polycomb-dependent H3K27me3 marks in *Dnmt*-tKO ES cells (*Saksouk et al., 2014*). Interestingly, we also observed H3K27me3 relocalization towards pericentromeric repeats in hypomethylated 2i+vitC ES cells. Similarly, the H3K36 demethylase KDM2B, which targets unmethylated CpG-rich sequences, was shown to recruit PRC1, potentially leading to H3K27me3 deposition through PRC2 recruitment (*Blackledge et al., 2014*; *Farcas et al., 2012*). Comparable mechanisms might be at play for the recruitment of H3K27me3 at hypomethylated transposons, involving BEND3, KDM2B and/or other methyl-sensitive DNA binding proteins.

Thus, based on previous observations, we posit that H3K27me3 invades the transposon space left unmarked by DNA methylation upon 2i+vitC conversion. Moreover, we provide evidence that the three possible chromatin configurations that the different transposon families adopt are further determined by H3K9me3 occupancy (*Figure 8B*). Mutual exclusion between H3K9me3 and H3K27me3 marks has been previously documented at gene promoters and pericentromeric repeats (*Mikkelsen et al., 2007*; *Peters et al., 2003*), but not at transposons. We found that category B transposons, which constantly maintain H3K9me3 marks throughout their entire length, do not acquire H3K27me3-based chromatin even though they lose DNA methylation. In contrast, category C transposons, exemplified by MERVL elements, become strongly enriched for H3K27me3 as H3K9me2/3 depletes during medium conversion. Finally, category A elements provide a striking illustration of the physical segregation of H3K9me3 and H3K27me3: as H3K9me3 constitutively marks the 5' end of this transposon category, only their 3' end is accessible to H3K27me3 deposition upon DNA methylation loss. The presence of H3K27me3 at the 3' end of transcription units has not been described before and its functional significance was unknown. Our transcriptome analysis of 2i+vitC-converted *Eed*-KO cells reveals that 3' end enrichment of H3K27me3 does not confer transcriptional repression of transposons of the A category: it may rather represent a passive response to the lack of DNA methylation and H3K9me3 at this position.

The main message conveyed by our work is that compensatory histone-based mechanisms ensure transposon silencing when DNA methylation-based control is alleviated in ES cells. We cannot rule out that other mechanisms-such as small RNA-based post-transcriptional repression-could also participate to transposon control. Importantly, genetic analyses globally confirmed the functionality of the chromatin patterns that we identified. In particular, H3K27me3-dependency was very well illustrated by the failure to repress several transposon families – in particular MERVL and LINE1– in *Eed*-KO ES cells undergoing medium-based DNA methylation loss. However, the transposon category B (IAPEz), which remains enriched for H3K9me3 throughout media conversion, gave complex, disparate phenotypes in the mutants of the different H3K9me3 pathways. While these elements failed to be repressed in *Setdb1* and *Trim28*-deficient ES cells, the complete suppression of IAPEz reactivation in *Suv39*-dko cells was unexpected. We suspect that alternative repressive processes likely

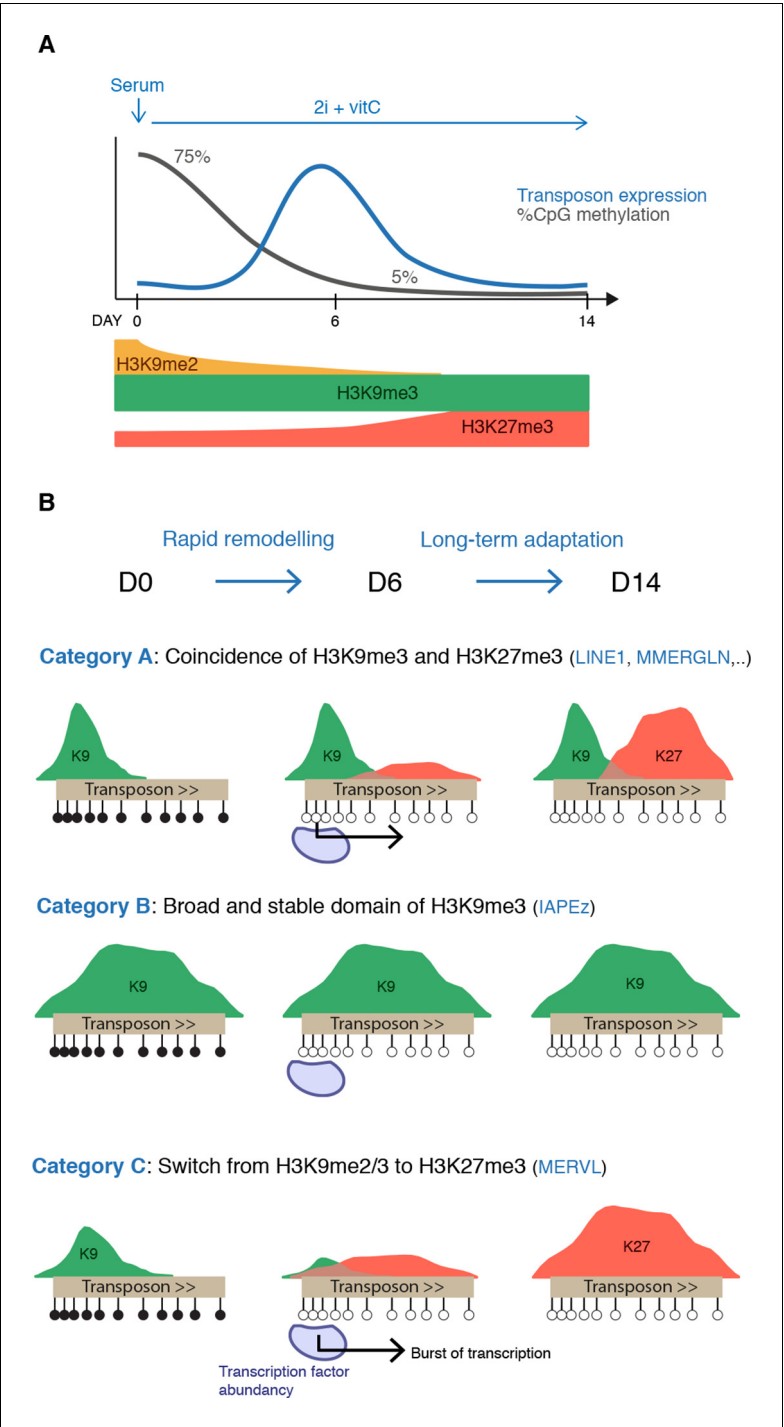

**Figure 8.** Model for the acquisition of H3K27me3 at transposons during genome-wide demethylation. (**A**) Summary of chromatin and transcriptional changes during conversion from serum to 2i+vitC. DNA methylation and H3K9me2 are rapidly erased, H3K9me3 remains stable and H3K27me3 increases. Transposon expression peaks at D6. (**B**) Model for the acquisition of H3K27me3 at transposons: upon loss of DNA methylation, H3K27me3 appears at GC-rich, H3K9me3 free-regions. Relative enrichments in H3K9me3 and H3K27me3 define three main types of repressive chromatin organization. Category A transposons are marked by H3K9me3 on their 5' end and gain H3K27me3 on their 3' region. Category B transposons are fully covered by H3K9me3 and do not gain H3K37me3. Category C transposons lose H3K9me2 and H3K9me3 and acquire H3K27me3 decoration on their full length. At D6 of 2i+vitC conversion, the abundance of pluripotent transcription factors and the loose chromatin environment likely contribute to the burst of transposon expression.

obscure IAPEz transcriptional responses to DNA methylation loss in this mutant. This is akin to *Dnmt*-tKO cells, which also exhibit global transposon repression. Thus, our analyses highlight the possible unexplained phenotypes of mutant cells that have adapted to long-term chromatin-based deficiencies.

Finally, one important point to raise is that the epigenetic switch from a DNA methylation-dependent to -independent mode of transposon silencing is not perfectly synchronized: ES cells experience an acute burst of transposon expression at D6 of medium conversion. At this time point, we showed that DNA methylation has been mostly erased but H3K27me3 patterns have not been established yet. Interestingly, the stability of H3K9me3 marks at category A and B transposons is not sufficient to ensure their continuous silencing upon conversion. This may imply that H3K9me3 readers are transiently deficient in this system. The lag between DNA methylation loss and subsequent implementation of histone-based repression could create an opportunistic window for transposon reactivation, provided that adequate transcription factors are available. Several studies have previously pointed out that transposons are enriched in pluripotency transcription factor binding motifs (*Kunarso et al., 2010*; *Wang et al., 2014*), in particular for NANOG and OCT4, and that upregulation of these transcription factors was sufficient to promote transposon expression (*Grow et al., 2015*).

We propose that the simultaneous disappearance of DNA methylation marks and increased availability of pluripotency activators create favorable conditions to transposon expression at D6 of serum to 2i+vitC conversion (*Figure 8B*). After a brief silencing release, functional repressive chromatin is recovered, in an H3K9me3 and/or H3K27me3-dependent manner. Notably, we repeatedly observed a peak of massive cell death of H3K27me3-deficient *Eed*-KO ES cells between D6 and D10 of medium conversion, when DNA methylation has mostly disappeared. This phenotype was even exacerbated in *Suv39h-Eed*-tKO ES cells, which did not survive after a week of medium conversion. These observations support the critical role for H3K27me3 in supplementing DNA methylation-based control in ES cells.

## Material and methods

### ES cell lines

J1 and *Dnmt*-tKO ES cells were a gift from M. Okano (*Tsumura et al., 2006*). E14 ES cells were kindly provided by E. Heard. WT TT2 and *Ehmt2*-KO ES cells (*Tachibana et al., 2002*), and *Eed*-KO (*Schoeftner et al., 2006*) (on a J1 background) were gifts from Y Shinkai and A Wutz, respectively. *Trim28+/-, Setdb1+/-* and *Suv39*-dKO were generated from J1 ES cells using CRISPR/Cas9 editing. Briefly, guide-RNAs specific to the target sequences were designed using the online CRISPR Design Tool (*Hsu et al., 2013* and *Supplementary file 2D*) and incorporated into the X330 backbone (*Cong et al., 2013*). Five millions J1 ES cells grown in serum were transfected with 1–3 μg of plasmid using Amaxa 4d nucleofector (Lonza) and plated at a low density. Individual clones were picked and screened by PCR; mutated alleles were confirmed by Sanger sequencing. *Suv39h*-dKO cells were obtained by creating a frame-shift in *Suv39h1* exon 4 and by deleting *Suv39h2* exon 4; *Trim28+/-* cells were generated by deleting exon 3; *Suv39h-Eed*-tKO were obtained by deleting *Eed* exon 6 in *Suv39h*-dKO cells; *Setdb1 +/-* were generated by creating a frameshift in exon 16.

### ES cell culture

ES cells were grown in two different media, serum and 2i, defined as follow. Serum: Glasgow medium (Sigma), 15% FBS (Gibco), 2 mM L-Glutamine, 0.1 mM MEM non essential amino acids (Gibco), 1 mM sodium pyruvate (Gibco), 0.1 mM β-mercaptoethanol, 1000 U/mL leukemia inhibitory factor (LIF, Miltenyi); 2i: 50% neurobasal medium (Gibco), 50% DMEM/F12 (Gibco), 2 mM L-glutamine (Gibco), 0.1 mM β-mercaptoethanol, Ndiff Neuro2 supplement (Milipore), B27 serum-free supplement (Gibco), 1000 U/mL LIF, 3 μM Gsk3 inhibitor CT-99021, 1 μM MEK inhibitor PD0325901. Vitamin C (Sigma) was added at a concentration of 100 ug/mL (*Blaschke et al., 2013*).

When in serum, J1, *Dnmt*-tKO, E14, and all CRISPR-generated mutant ES cells were grown in feeder-free conditions on gelatin-coated plates. TT2, *Ehmt2*-KO were cultured on a monolayer of mitomycin C-treated mouse embryonic fibroblasts. ES cells were passaged with TrypLE Express

Enzyme (Life Technologies, Carlsbad, CA). All 2i ES cells were grown in gelatin-coated plates and passaged every two or three days with Accutase (Life Technologies).

Trichostatin A was added for 24 hr at concentration of 25 or 50 ng/mL

Mycoplasma-free status was assessed using VenorGeM Classic mycoplasma detection kit (Minerva Biolabs).

## DNA methylation analyses

Genomic DNA was isolated using the GenElute Mammalian Genomic DNA Miniprep Kit (Sigma) with RNase treatment. Global CpG methylation levels were assessed using LUminometric Methylation Assay (LUMA) as described previously (*Karimi et al., 2011a*; *Richard Pilsner et al., 2010*). Briefly, 500 ng of genomic DNA was digested with *MspI/EcoRI* and *HpaII/EcoRI* (NEB) in parallel reactions. *HpaII* is a methylation-sensitive restriction enzyme and *MspI* is its methylation insensitive iso-schizomer. *EcoRI* is included as an internal reference. The overhangs created by the enzymatic digestion were quantified by Pyrosequencing (PyroMark Q24, Qiagen) with the dispensation order: GTGTGTCACACAGTGTGT. Global CpG methylation levels were calculated from the peak heights at the position 7,8,13,14 as follows: $1-\mathrm{sqrt}([p8*p14/p7*p13]_{HpaII} /[p8*p14/p7*p13]_{MspI})$

CpG methylation at specific loci was assessed by bisulfite-pyrosequencing using the Imprint DNA modification Kit (Sigma) for conversion. PCR and sequencing primers (*Supplementary file 2D*) were designed with the PyroMark Assay Design Software and quantification of DNA methylation was performed according to the recommended protocol.

Whole-Genome Bisulfite Sequencing libraries were prepared from 50ng of bisulfite-converted genomic DNA using the EpiGnome/Truseq DNA Methylation Kit (Illumina) following the manufacturer instructions. Sequencing was performed in 100 pb paired-end reads at a 30X coverage using the Illumina HiSeq2000 platform (*Supplementary file 1*).

## RNA expression analyses

Total RNA was extracted using Trizol (Life Technologies). cDNAs were prepared after DNase treatment (Turbo DNase, Ambion) using random priming with Superscript III (Life Technologies). Real-time quantitative PCR was performed using the SYBR Green Master Mix on the Viia7 thermal cycling system (Applied Biosystem). Relative expression levels were normalized to the arithmetic mean of the housekeeping genes *Gapdh* and *Rplp0* and to WT-D0 with the ΔΔCt method. Primers are given in *Supplementary file 2D*.

Nanostring nCounter quantification was performed using 100ng of total RNA per sample on a custom expression Codeset (target sequences in *Supplementary file 2D*). *Actin*, *Ppia*, *Gapdh* and *Rplp0* were used for normalization. Data are presented as the fold change compared to WT-D0. Expression data for the different mutants are presented next to WT data that were processed on the same Nanostring run. The same WT data can be used in several figures. When necessary and in order to calculate mean and standard error of the mean between replicates every two days, we extrapolated linearly the expression value of a given day using data of immediately adjacent time points (for both RT-qPCR and Nanostring).

RNA-seq libraries were prepared from 500ng of DNase-treated RNA with the TruSeq Stranded mRNA kit (Illumina). Sequencing was performed in 100pb paired-end reads using the Illumina HiSeq2000 platform (*Supplementary file 1*).

## Chromatin Immunoprecipitation

Cells were cross-linked directly in culture plates with 1% formaldehyde (culture medium supplemented with 1% formaldehyde, 0.015 M NaCl, 0.15 mM EDTA, 0.075 mM EGTA, 15 mM Hepes pH 8). After quenching with 0.125 M glycine, cells were washed in PBS and pelleted. Cells were then incubated at 4°C for 10 min in buffer 1 (Hepes-KOH pH 7.5 50 mM, NaCl 140 mM, EDTA pH 8.0 1 mM, glycerol 10% NP-40 0.5%, Triton X-100 0.25% and the protease inhibitors: PMSF 1 mM, Aprotinin 10 µg/ml, leupeptin 1 µg/ml and pepstatin 1 µg/ml), then at room temperature for 10 min in buffer 2 (NaCl 200 mM, EDTA pH 8.0 1 mM, EGTA pH 8.0 0.5 mM and 10 mM Tris pH 8 and the same protease inhibitors as buffer 1) and finally resuspended in buffer 3 (EDTA pH 8.0 1 mM, EGTA pH 8.0 0.5 mM, Tris pH8 10 mM, N-lauroyl-sarcosine 0.5%; protease inhibitors as buffer 1). Chromatin was sonicated with a Bioruptor (Diagenode) to reach a fragment size around 200 bp. Chromatin

corresponding to 10 µg of DNA was incubated overnight at 4°C with 3–5 µg of antibody in incubation buffer (buffer 3 supplemented with 0.5 volume of 3% Triton, 0.3% sodium deoxycholate, 15 mM EDTA; protease inhibitors). A fraction of chromatin extracts (5%) were taken aside for inputs. Antibody-bound chromatin was recovered using Protein G Agarose Columns (Active Motif). Briefly, the antibody-chromatin mix was incubated in the column for 4 hr, washed eight times with modified RIPA buffer (Hepes pH7.6 50 mM, EDTA pH 8.0 10 mM, sodium deoxycholate 0.7%, NP-40 1%, LiCl 500 mM, PMSF 1 mM, 1 µg/ml leupeptin and 1 µg/ml pepstatin), and washed one last time with TE-NaCl (50mM Tris pH 8.0, 10 mM EDTA, 50 mM NaCl). Chromatin was eluted with pre-warmed TE-SDS (50mM Tris pH 8.0, 10 mM EDTA, 1% SDS). ChIP-enriched sample and inputs were then reverse cross-linked at 65°C overnight and treated with RNase A and proteinase K. DNA was extracted with phenol/chloroform/isoamyl alcohol, precipitated with glycogen in sodium acetate and ethanol and finally resuspended in TE. Enrichment compared to input was analyzed by qPCR. A quantity of 20 ng of ChIP- or input-DNA were used for ChIP-seq. Remaining large DNA fragments were first eliminated using SPRIselect beads (Beckman Coulter) and libraries were prepared using the TruSeq ChIP Sample Prep kit (Illumina). Sequencing was performed in 50pb paired-end reads using the Illumina HiSeq2000 platform (*Supplementary file 1*). qPCR primers and antibodies are listed in *Supplementary file 2D,E*.

## Western blotting

To prepare total protein extracts, cells were resuspended in BC250 lysis buffer (25 mM Tris pH 7.9, 0.2 mM EDTA, 20% Glycerol, 0.25 M KCl and protease inhibitor coktail from Roche), sonicated and centrifuged to pellet debris. To prepare nuclear protein extracts, cells were incubated for 10 min on ice in buffer A (Hepes pH 7.9 10 mM, MgCl2 5 mM, Sucrose 0.25 M, NP40 0.1%, DTT 1mM and protease inhibitors) and centrifuged. The pellet was resuspended in buffer B (Hepes pH 7.9 25 mM, glycerol 20%, MgCl2 1.5 mM, EDTA 0.1 mM, NaCl 700 mM, DTT 1 mM and protease inhibitors), sonicated and centrifuged to pellet debris. Total and nuclear proteins were quantified by Bradford assay. Proteins (10–20 µg per gel lane) were separated by electrophoresis in 8–15% poly-acrylamide gels and transferred onto nitrocellulose membranes using the Trans-Blot turbo transfer system (Biorad). After incubation with primary antibodies and HRP-conjugated secondary antibodies, signal was detected using ECL prime kit (Amersham) and ImageQuant Las-4000 mini biomolecular Imager. Antibodies are listed in *Supplementary file 2E*.

## Immunofluorescence

Cells were harvested with Trypsin or Accutase, resuspended in PBS and plated for 10 min on Poly-L-Lysine-coated glass cover slips. Cells were first fixed with 3% paraformaldehyde for 10 min at room temperature, then rinsed three times with PBS and permeabilized for 4 min with 0.5X Triton on ice. After blocking in 1% BSA for 15 min, samples were incubated at room temperature for 40 min with primary antibodies, 45 min with secondary antibodies and 3 min in 0.3 µg/mL DAPI. Slides were mounted with Prolong Gold mounting media (Invitrogen). Images were obtained with an Upright Widefield microscope (Leica) or a Zeiss LSM700 inverted confocal microscope. Quantification of immunofluorescence intensity in individual cells was performed using custom ImageJ and R scripts. Between 2000 and 5000 cells were analyzed per sample. Antibodies are listed in *Supplementary file 2E*.

## Metaphase spreading

Cells were cultured for two hours with 0.04 µg/mL colchicine and harvested by trypsinization. Cell pellets were incubated in hypotonic buffer (15% FBS in water) for 7 min at 37°C and fixed with 66% acetic acid/33% ethanol. After centrifugation, cells were resuspended in 1.5 mL fixative and dropped from ~1 m height onto glass slides. Slides were dried and DNA was stained with DAPI. Chromosomes were counted with an Upright Widefield microscope (Leica). Around 20 cells were analyzed per cell line.

## Quantification of transposon genomic copy number

Absolute copy numbers of IAP and LINE1 were calculated by qPCR by establishing standard curves plotting absolute Ct values of genomic DNA against serial dilutions of PCR targets cloned into the pCR2.1-TOPO vector (Life Technologies), as described in *Zamudio et al., 2015*.

## Reconstruction of repeatMasker

As described in *Bailly-Bechet et al., 2014*, a dictionary was constructed for LTR retrotransposons that associated elements corresponding to the internal sequence and those corresponding to LTR sequences. With the latter and the RepeatMasker database, fragments of transposable elements corresponding to the same copy were merged. Divergence, deletion and insertion percentages were recalculated from RepeatMasker and an integrity score for each transposon were calculated as follow: score = 1-average(%divergence,% deletions,% insertions)

## WGBS data analysis

Whole-genome bisulfite sequencing reads generated in this study or recovered from available data-sets were treated as follow. The first eight base pairs of the reads were trimmed using FASTX-Toolkit v0.0.13 (http://hannonlab.cshl.edu/fastx_toolkit/index.html). Adapter sequences were removed with Cutadapt v1.3 (https://code.google.com/p/cutadapt/) and reads shorter than 16 bp were discarded. Cleaned sequences were aligned onto the Mouse reference genome (mm10) using Bismark v0.12.5 (*Krueger and Andrews, 2011*) with Bowtie2-2.1.0 (*Langmead and Salzberg, 2012*) and default parameters. Only reads mapping uniquely on the genome were conserved. Methylation calls were extracted after duplicate removal. Only CG dinucleotides covered by a minimum of 10 reads were conserved for the rest of the analysis.

The R-package Methylkit v0.9.2 (*Akalin et al., 2012*) was used to provide Pearson's correlation scores between samples. To analyze the distribution of CpG methylation in different genomic compartments, the mouse genome was divided into different partitions. The RefSeq gene annotation and the RepeatMasker database were downloaded from UCSC table browser and used for transcript and repeat annotations, respectively. Promoters were defined as the -1 kb to +100 pb region around transcription start sites. CpG islands (CGIs) were defined as in *Illingworth et al., 2010*. Intergenic partitions were defined as genomic regions that did not overlap with promoters, CGI, exons, introns or repeats. Whole-genome mapping of CpG methylation was then intersected with the different genomic compartments using Bedtools (*Quinlan and Hall, 2010*).

Average CpG methylation on individual transposons was extracted from RepeatMasker with Bedtools, average CpG methylation in the different transposon families was calculated and plotted using R. Heatmap for average CpG methylation in Imprinted control regions (ICRs) was generated similarly after retrieving ICR genomic coordinates from the WAMIDEX database (*Schulz et al., 2008*). Residually methylated regions (RMRs) in 2i+vitC samples were identified using the MethPipe pipeline (*Song et al., 2013*) with default parameters. RMRs located less than 1 kb from each others were concatenated.

## RNA-seq data analysis

In order to quantify gene expression, Paired-end 2x100 bp reads were mapped onto mm10 using Tophat v2.0.6 and RefSeq gene annotation (*Kim et al., 2013*) allowing five mismatches. Gene-scaled quantification was performed with HTSeq v0.6.1 (*Anders et al., 2014*).

In order to quantify transposon expression, reads mapping to ribosomal RNA (rRNA) sequences (GenBank identifiers: 18S NR_003278.3, 28S NR_003279.1, 5S D14832.1, 5.8S K01367.1) were first removed with Bowtie v1.0.0 allowing three mismatches. The rRNA-depleted libraries were then mapped onto mm10 using Bowtie v1.0.0 allowing zero mismatch and 10000 best alignments per read. Exonic reads were removed. In order to count reads mapping to transposable elements, reads were weighted by the number of mapping sites and each library was intersected with the reconstructed RepeatMasker annotation, conserving only reads overlapping at least at 80% with a given transposon.

For each library, read counts for genes and transposons were combined into a single table. TMM normalization from the edgeR package v3.6.2 (*Robinson and Oshlack, 2010*) was first applied. As described in the guideline of limma R-package v3.20.4, normalized counts were processed by the

voom method (*Law et al., 2014*) to convert them into log2 counts per million with associated precision weights. The differential expression was estimated with the limma package. Genes and transposons were called differentially expressed when two criteria were met: 1) the fold-change between two conditions was higher than four and two, respectively, and 2) the adjusted p-value using the Benjamini Hochberg procedure was below 0.05.

For the analysis of RNA-seq libraries with uniquely mapped reads, the mapping was performed as previously with Bowtie v1.0.0, except that only uniquely mapping reads were conserved. Read counts on individual reconstructed element were quantified using HTSeq v0.6.1. Only elements with at least 10 reads in at least one sample were conserved for further analysis and read counts were subsequently normalized by the library size. Normalized read counts for individual elements belonging to different families were then plotted using custom R script. Tracks were created using HOMER software v4.7 (*Heinz et al., 2010*).

In order to identify and characterize chimeric transcripts, reads were mapped onto mm10 using Tophat v2.0.6, without providing a gene annotation. Cufflinks v2.2.1 (*Trapnell et al., 2010*) was used to reconstruct the transcriptome and quantify the different isoforms. Transcripts were considered chimeric when the first exon overlapped with a transposon annotated in Repeatmasker and one of the other exon was annotated in RefSeq.

### ChIP-seq data analysis

Paired-end 2x50bp reads were mapped onto mm10 using Bowtie v1.0.0 allowing 3 mismatches. Reads mapping to multiple locations were randomly allocated. Duplicate reads were removed using Picard v1.65 (http://broadinstitute.github.io/picard/). Tracks were created using HOMER software v4.7 (*Heinz et al., 2010*) and Peak calling was performed with MACS2 v2.0.10 (*Zhang et al., 2008*) using the broad option and a 5% FDR threshold. Detected peaks were annotated using RefSeq and RepeatMasker databases. In order to construct the heatmap and the scatter plots, the total number of read counts for every annotated transposable element was computed using Bedtools and the reconstructed RepeatMasker annotation. Enrichment was normalized by the size of the element and Input data. Metaplots for average enrichment and GC content on and around different transposons were obtained using HOMER V4.7. Only full-length (>6 kb) and intact (integrity score >0.8) elements were used for the metaplots.

## Acknowledgements

We thank the members of DB's laboratory, especially M Greenberg, J Barau and T Chelmicki for critical input and experimental help. We thank M Okano, A Wutz, Y Shinkai and E Heard for the gift of mouse ES cells lines; E Heard, G Almouzni, R Margueron, B Cullen and A Bortvin for antibodies. We acknowledge the PICTIBiSA@BDD for microscopy; the Institut Curie NGS platform supported by the ANR-10-EQPX-03 and ANR10-INBS-09-08 grants and the Canceropôle Ile-de-France for high-throughput sequencing; the Genomic Platform for Nanostring ncounter analysis. DB's laboratory is part of the Laboratoire d'Excellence (LABEX) entitled DEEP (11-LBX0044). This research was supported by grants from the European Research Council (ERC) and ANR ('ABS4NGS'-ANR-11-BINF-0001). MW is recipient of a PhD fellowship from the Ecole Polytechnique.

## Additional information

### Funding

| Funder | Grant reference number | Author |
|---|---|---|
| European Research Council | Consolidator | Déborah Bourc'his |
| Agence Nationale de la Recherche | 11-LBX0044 | Déborah Bourc'his |

The funders had no role in study design, data collection and interpretation, or the decision to submit the work for publication.

## Author contributions

MW, Performed all the experiments, at the exception of the immunofluorescence studies, Conducted bioinformatics analyses, Conception and design, Acquisition of data, Analysis and interpretation of data, Drafting or revising the article; AT, Conducted bioinformatics analyses, Analysis and interpretation of data; RPP, Performed the immunofluorescence studies, Acquisition of data; DB, Conception and design, Analysis and interpretation of data, Drafting or revising the article

# Additional files

## Supplementary files

• Supplementary file 1. WGBS, RNA-seq and ChIP-seq statistics

• Supplementary file 2. (A) Number and percentage of active transposable elements at D0, D6 and D13 during conversion from serum to 2i+vitC. Elements were considered as 'active' when they were covered by at least 10 uniquely mapped reads at one of the time point. Percentages represent the proportion of active copies relative to the total number of elements in a given family, as estimated from the reconstructed version of RepeatMasker. For ERVs, solo-LTRs were excluded and numbers represent only elements containing internal sequences. (B) Gene Ontology enrichment analysis for genes specifically up-regulated between D0 and D6. Gene ontology analysis for biological processes and molecular functions was performed for the 156 genes that were significantly up-regulated between D0 and D6 but with no significant differences between D0 and D13. The ten most significant terms are shown. (C) List of chimeric transcripts identified during conversion from serum to 2i +vitC. The 30 genes with the highest number of chimeric reads at D6 are ranked here. Numbers represent the absolute read count at the junction between the transposon (first exon) and the second exon of the gene, and the normalized read count of the whole transcript in RPKM. (D) Primer and sequence list. (E) Antibody list.

## Major datasets

The following datasets were generated:

| Author(s) | Year | Dataset title | Dataset URL | Database, license, and accessibility information |
|-----------|------|---------------|-------------|--------------------------------------------------|
| Walter M, Teissandier A, Pérez Palacios R, Bourc'his D | 2015 | An epigenetic switch ensures transposon repression upon acute loss of DNA methylation in ES cells | http://www.ncbi.nlm.nih.gov/geo/query/acc.cgi?acc=GSE71593 | Publicly available at the NCBI Gene Expression Omnibus (Accession no: GSE71593). |

The following previously published datasets were used:

| Author(s) | Year | Dataset title | Dataset URL | Database, license, and accessibility information |
|-----------|------|---------------|-------------|--------------------------------------------------|
| Seisenberger S, Andrews S, Krueger F, Arand J, Walter J, Santos F, Popp C, Thienpont B, Dean W, Reik W | 2012 | The dynamics of genome-wide DNA methylation reprogramming in mouse primordial germ cells | http://www.ebi.ac.uk/ena/data/view/ERP001953 | Publicly available at the EBI European Nucleotide Archive (Accession no: ERP001953). |
| Marks H, Habibi E, Brinkman AB, Arand J, Kroeze LI, Kerstens HH, Matarese F, Lepikhov K, Gut M, Brun-Heath I, Hubner NC, Benedetti R, Altucci L, Jansen JH, Walter J, Gut IG, Stunnenberg HG | 2013 | Whole-genome bisulfite sequencing of two distinct interconvertible DNA methylomes of mouse embryonic stem cells | http://www.ncbi.nlm.nih.gov/geo/query/acc.cgi?acc=GSE41923 | Publicly available at the NCBI Gene Expression Omnibus (Accession no: GSE41923). |

| | | | | |
|---|---|---|---|---|
| Brinkman AB, Gu H, Bartels SJ, Zhang Y, Matarese F, Simmer F, Marks H, Bock C, Gnirke A, Meissner A, Stunnenberg HG | 2012 | Sequential ChIP-bisulfite sequencing enables direct genome-scale investigation of chromatin and DNA methylation cross-talk | http://www.ncbi.nlm.nih.gov/geo/query/acc.cgi?acc=GSE28254 | Publicly available at the NCBI Gene Expression Omnibus (Accession no: GSE28254). |

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
