## [Decision Letter]

Thank you for submitting your work entitled "An epigenetic switch ensures transposon repression upon dynamic loss of DNA methylation in ES cells" for consideration by *eLife*. Your article has been reviewed by three peer reviewers, and the evaluation has been overseen by Anne Ferguson-Smith (Reviewing Editor) and Fiona Watt as the Senior Editor.

Two of the three reviewers involved in review of your submission have agreed to reveal their identity: Jorn Walter (reviewer 1) and Todd MacFarlan (reviewer 2). For clarification, reviewer 3 remains anonymous.

The reviewers have discussed the reviews with one another and the Reviewing editor has drafted this decision to help you prepare a revised submission.

Below, you will find a list of the comments from the reviewers that have not been modified because we believe that you will find it useful to consider all the points. As you will see, there is consistency between the reviewers with regard to appropriate statistical analysis. This may also address some of the issues of interpretation that are raised. We expect you to conduct a thorough statistical analysis of the data in a revised manuscript. Reviewer 2 has provided a comprehensive list of detailed comments – please address them all in a revised manuscript. These are probably unlikely to involve additional experimentation but will require further analysis and some changes to the text in several places. Reviewer 3 raises an interesting point about heterogeneity during the mid-treatment timepoint. Please consider this in your revised manuscript.

*Reviewer #1:*

The paper by Walter et al. describes the fate of repressive/heterochromatic modifications and their effect on transcriptional activity of retrotransposable elements in mouse ES cells upon gradual loss of DNA-methylation. The paper focuses on the analysis of altered histone modification patterns following a global and very general (i.e. genome wide) 2i+VitC induced DNA-demethylation. The major findings are: i) A burst of initial DNA-demethylation causes distinct patterns of initial transcriptional derepression which ii) appears to be differentially compensated by an element specific gain of heterochromatic marks leading iii) to a partial re-silencing. The paper provides a very deep and differentiated view on transposon control in murine ES cells. The data provides a detailed analysis of the complex dynamics and interplay of different layers of heterochromatic marks. The main finding is that different classes of (retro-) transposons are controlled by partially redundant but distinct networks of heterochromatic histone marks in ESCs. In addition the paper provides an excellent resource of unique DNA-methylation and ChIP-Seq data.

The paper is very well structured and well written. The reader is nicely guided through a complex set of consecutive and logically ordered analyses. The paper compiles a huge amount of experimental data dissected for a wide spectrum of repetitive element classes in the mouse and provides an excellent resource of information for many researchers in the field of stem cell biology and epigenetics. The straight forward bioinformatic comparative analyses culminates in the definition of three epigenetic "ground" states (A, B and C). Using (and generating) genetic manipulations the authors also aim to unravel functional links between the heterochromatic control and enzyme contributions. These analyses are well performed but do not provide many novel insights in the biology of transposon control.

Overall the excellent paper provides thoroughly interpreted data adding a number of interesting new aspects to previous findings on the role of heterochromatic control in ESCs. In summary this a very nice piece of work with a high impact for epigenetics of stem cells.

Major comment: A deeper evaluation of the data, e.g. using statistical learning and/or training methods would have enhanced the paper instead of using groupwise classifications.

*Reviewer #2:*

In this manuscript from Deborah Bourc'his and colleagues, the authors used a defined media culture switch (serum containing medium to 2i + vitamin C) in ESCs to rapidly demethylate the genome to test the underlying hypothesis that DNA methylation might actually be required for transposon silencing in ESCs. This goes against the current dogma that suggests that DNA methylation is dispensable in ESCs (based on the use of Dnmt 3KO ESCs.) The authors nicely demonstrate that there is a phase of transposon expression following the rapid demethylation, followed by resilencing of transposons. Using RNA-seq, ChIP-seq, and bioinformatics approaches, they identify different classes of transposons based on the mechanisms of re-silencing, which includes polycomb-dependent H3K27me3. This paper is logical, very clearly written, and very nicely carried out, however there are a few concerns that should be addressed before I can recommend publication.

1) In the Discussion, the authors state that their work "revises the previous assumption that DNA methylation is dispensable for transposon silencing in ES cells". However, the observed upregulation of transposons (e.g. IAPez, 3-fold, MERVL, 2-fold) is rather weak upon demethylation (as compared to what is observed in KAP1 and ESET KO ES cells (IAP) or LSD1 KOs (MERVL)). Furthermore, as the authors comment themselves, the upregulation of pluripotency transcription factors upon demethylation could be responsible for the observed ERV reactivation. The authors also show that more than 3000 other genes are up- or down-regulated during conversion; many of those genes might influence transposon expression/repression. I cannot think of a way to exclude these potential indirect effects but the authors should mention that some the observed transposon upregulation might not be directly caused by loss of DNA methylation at these elements and be more careful in implying that DNA methylation is generally required for transposon silencing in ES cells. It rather seems that some ERVs that are not marked by H3K9me3 (e.g. MERVL) require both DNA methylation and H3K27me3 to remain in a repressed state in certain culture conditions whereas DNA demethylation has very little (maybe merely indirect) effects on ERVs repressed by the KAP1/ESET system.

2) ESCs, whether cultured using classical media conditions or those that induce demethylation are an imperfect model system at best for what occurs during development. Thus, these current results beg the question what is actually occurring in preimplantation embryos, where many transposons are demethylated. Can the authors demonstrate that H3K27me3 is enriched on L1 or MERVL elements as they are de-methylated during normal development?

3) The authors speculate that H3K27me3 is established at demethylated transposons by mechanisms that recognize unmethylated CpG sites. The authors should indicate the CpG content in Figure 5 to test such an association. Also, it is possible that increased transcription results in H3K27me3 enrichment. It would be interesting to plot the transcriptional activity of all uniquely mappable transposons against their H3K27me3 enrichment to test this possibility. If possible, the authors could also try to analyze H3K27me3 at fragmented transposons (e.g. ERVs without LTRs or LTRs without internal regions) to define the regions that are essential for H3K27me3 targeting.

4) Strangely absent from the experiments/discussion about MERVL elements is the function of histone deacetylases (which can be inhibited pharmacologically with TSA) and LSD1 (KO ESC lines are available from multiple labs), which have both been shown to play an important role in MERVL silencing in ESCs (Macfarlan and Pfaff, Genes and Development 2011). Are these factors required for the re-silencing following the rapid DNA methylation?

5) The experiment treating Dnmt 3KO ESCs with 2i and vit C is a nice experiment and important control. However the results shown in the supplementary figure are not that convincing, since the J1 controls look nothing like the ESCs used in the main figures. There are no statistics performed on these figures which is a bit concerning for the overall interpretation of the methodology.

6) I disagree with the assertion (made several times) that MERVL elements "switch from H3K9me3 to H3K27me3-regulated." MERVL elements are mostly devoid of H3K9me3 but instead have high levels of H3K9me2, which is lost upon LSD1 or G9a deletion (Macfarlan and Pfaff, G and D 2011, Maksakova and Lorincz, Epigenetics and chromatin 2013, also Figure 4). This is consistent with MERVLs being unresponsive to SETDB1 deletion. Thus this third category of transposons should be changed to reflect the data presented in the manuscript and the previous publications.

7) At day 6, IAP elements are derepressed but are still somewhat methylated. Can the authors address the methylation status of the expressed IAPs? This could perhaps be achieved by ChIP'ing first with H3K9me3, or with active marks like RNAPII or AcH3 marks, followed by bisulfite sequencing analysis.

*Reviewer #3:*

Walter et al in their manuscript demonstrate interesting epigenetic changes at transposable elements using a chemically-induced system for hypomethylation of ES cells. The ability to measure temporal changes in this system allowed the authors to observe a transient upregulation of transposable element expression associated with a change in histone modifications at these sites. Specifically, across all subfamilies of repetitive elements, the authors observed three patterns of histone modification changes during hypomethylation: (1) gain of both H3K9me3 and H3K27me3, (2) exclusive H3K9me3 occupancy, and (3) a switch from H3K9me3 to H3K27me3 occupancy. Notably, subfamilies within the first group, including LINEs and subfamilies of ERV1 and ERVK, showed heterogeneity of behavior and a spatial segregation of H3K9me3 and H3K27me3 with H3K9me3 preferentially located at the 5' end and H3K27me3 located at the 3' end. The authors lastly aimed to test ideas about histone repression of transposable elements using isogenic mES cell knockouts of chromatin modifying enzymes, Suv39h, Kap1, and Eed, responsible for H3K9me3 and H3K27me3. The results show a complexity between chromatin modifiers with some consistent and others unexplained by the data. Overall, this study provided an interesting observation of two repressive modifications implicated in repression of transposable elements in the place of DNA methylation in order to understand regulation of these elements during developmental stages of hypomethylation.

However, a major pitfall of the current study is that all the analyses (DNA methylation, gene expression and histone modifications) are done at the cell population level during the time course of 13 days of 2i+vitC treatment. The authors have to consider the issue of cell heterogeneity in mid of 2i+vitC treatment (e.g. D6), which may lead to the different interpretations of their data. For example, it has been demonstrated that DNA hypomethylation-mediated de-repression of retrotransposons is associated with the degree of cell differentiation state (e.g. Hutnick et al. PMID 20404320, to be cited in this manuscript). When ESCs are grown in serum conditions, the 2i+vitC treatment would impose a selection, leading to the transient differentiation of a subset of ESCs in the time course. Therefore, as recently done by Grow EJ et al. (Nature 2015), the authors should perform immunocytochemistry or FISH analysis of ESCs to rule out the burst of transient de-repression of retrotransposons is only taking place in a subset of differentiated cells after 6days of 2i+vitC treatment. Alternatively, multiplex single cell RT-PCR assays of ESCs with transposons and cell differentiation markers may also resolve this critical issue.

1) In the section of "Transposons undergo a biphasic mode of regulation upon serum to 2i+vitC conversion". This section needs to be resolved by IHC or multiplex single cell RT-PCR to determine whether the bursting of retrotransposon expression is only in a subset of heterogeneous cells during the time course of 2i+vitC conversion.

2) In Figure 3, the authors showed RNA-seq analysis of differential gene expression during the 2i+vitC conversion. It would be obvious to the authors whether they have detected the up-regulation of differentiation genes such as ectoderm, mesoderm and endoderm marker genes in D6 samples. In fact, the transient upregulation of Tbx3 gene is consistent with this possibility (Figure 3).

3) Statistical analysis should be performed for many data points. For example, in Figure 6, are the differences of expression in different time points statistically significant?

4) In Figure 2, it is mentioned that Dnmt-tKO cells did not show transposon transcriptional reactivation, which is in strong contrast to the 2i+vitC. The explanation given is that long-term compensatory mechanisms have repressed these sites. However, there is no established mechanism. If H3K9me3 is not increased in Dnmt-tKO, does H3K27me3 increase in Dnmt-tKO under 2i-vitC conditions? It is worthwhile to study or discuss the discrepancy in more detail.

---

## [Author Response]

Reviewer #1:

*[…] Overall the excellent paper provides thoroughly interpreted data adding a number of interesting new aspects to previous findings on the role of heterochromatic control in ESCs. In summary this a very nice piece of work with a high impact for epigenetics of stem cells.*

*Major comment: A deeper evaluation of the data, e.g. using statistical learning and/or training methods would have enhanced the paper instead of using groupwise classifications.*

We would like to thank the referee for praising our work. We understand that he wished we had developed some more complex mathematical analyses based on our findings but we have to acknowledge that this is not our area of expertise. But statistical analyses have been now systematically added, using Welch’s t-test (unequal variances t-test).

Reviewer #2:

*[…] This paper is logical, very clearly written, and very nicely carried out, however there are a few concerns that should be addressed before I can recommend publication. 1) In the Discussion, the authors state that their work "revises the previous assumption that DNA methylation is dispensable for transposon silencing in ES cells". However, the observed upregulation of transposons (e.g. IAPez, 3-fold, MERVL, 2-fold) is rather weak upon demethylation (as compared to what is observed in KAP1 and ESET KO ES cells (IAP) or LSD1 KOs (MERVL)). Furthermore, as the authors comment themselves, the upregulation of pluripotency transcription factors upon demethylation could be responsible for the observed ERV reactivation. The authors also show that more than 3000 other genes are up- or down-regulated during conversion; many of those genes might influence transposon expression/repression. I cannot think of a way to exclude these potential indirect effects but the authors should mention that some the observed transposon upregulation might not be directly caused by loss of DNA methylation at these elements and be more careful in implying that DNA methylation is generally required for transposon silencing in ES cells. It rather seems that some ERVs that are not marked by H3K9me3 (e.g. MERVL) require both DNA methylation and H3K27me3 to remain in a repressed state in certain culture conditions whereas DNA demethylation has very little (maybe merely indirect) effects on ERVs repressed by the KAP1/ESET system.*

The referee disagrees with our conclusion that DNA methylation is required for transposon repression in ES cells, based on the following arguments:

A) Levels of transposon reactivation are rather weak upon medium-induced DNA demethylation of J1 ES cells. We would like to point the fact that the upregulation is more important in E14 cells, with levels ranging from 5 to 15 fold (Figure 2—figure supplement 1). Moreover, it is likely that these limited reactivation levels reflect the fact that while DNA methylation disappears, chromatin-based controls have already started being implemented at D6. In other words, the measured steady-state levels would be the sum of DNA methylation disappearance and gain of chromatin-based repression.

B) The upregulation could be due to a greater availability of pluripotency-related transcription factors. Indeed, to be transcribed, transposons need to be bound by transcription factors. This is a point we are aware of and that we have stressed in the Discussion by suggesting that pluripotency-related transcription factors, whose expression also peaks upon medium conversion, could be involved. However, the absence of reactivation in *Dnmt-*tKO ES cells upon serum to 2i+vitC switch, especially for IAPs, suggests that the greater availability of certain transcription factors upon medium conversion cannot solely account for the burst of expression.

C) DNA methylation has little effect on ERVs repressed through the KAP1/ESET system.

We suggest an alternative explanation: the lack of reactivation of IAPs in *Dnmt-*tKO cells (and *Suv39h-*dKO cells) rather implies that DNA methylation is involved in IAP repression, but that *Dnmt-*tKO cells have adapted to compensate for its absence, through polycomb repression.

*2) ESCs, whether cultured using classical media conditions or those that induce demethylation are an imperfect model system at best for what occurs during development. Thus, these current results beg the question what is actually occurring in preimplantation embryos, where many transposons are demethylated. Can the authors demonstrate that H3K27me3 is enriched on L1 or MERVL elements as they are de-methylated during normal development?*

Indeed, our findings open the path for investigating H3K27me3 enrichment at transposons during developmental periods of DNA methylation loss. However, such experiments are not trivial to perform, as they require collecting sufficient amounts of relevant material, i.e. primordial germ cells (PGCs) or preimplantation embryos for ChIP. This is certainly something we could consider doing, as a follow up of this work, but this would represent a whole study per se.

Nonetheless, it is worth mentioning that the global chromatin reorganization that we observed in our culture-based ES cell system has also been reported by immunological assays in PGCs (Hajkova et al., 2008; Seki et al., 2007). On the same point, although this ChIP-seq study presents some caveats, Liu et al., 2014 recently reported a co-occurrence of H3K27me3 and H3K9me3 marks at transposons in hypomethylated E13.5 PGCs. These results suggest that our system provides a relevant model for in vivoevents of DNA methylation reprogramming. As the in vivo biological relevance of our findings is an interesting point to discuss, this comment has now been added in the Discussion.

*3) The authors speculate that H3K27me3 is established at demethylated transposons by mechanisms that recognize unmethylated CpG sites. The authors should indicate the CpG content in Figure 5 to test such an association.*

The GC-content of transposons was already shown in Figure 5—figure supplement 2. We prefer not to overcharge the primary figure with data that we discuss only at the end of the manuscript.

*Also, it is possible that increased transcription results in H3K27me3 enrichment. It would be interesting to plot the transcriptional activity of all uniquely mappable transposons against their H3K27me3 enrichment to test this possibility.*

This is indeed an appealing possibility to investigate. Following the referee’s comment, we attempted to perform such analysis, but we are sorry to say we were unsuccessful in this endeavor. In RNA-seq data, a single polymorphism into a highly transcribed transposon will be sufficient to map many reads and prove that this element is expressed. In contrast, in ChIP-seq datasets, reads mapping to a specific polymorphism are in low numbers; moreover, they can only inform about the local chromatin state but this cannot be extrapolated to the rest of the transposon sequence.

If possible, the authors could also try to analyze H3K27me3 at fragmented transposons (e.g. ERVs without LTRs or LTRs without internal regions) to define the regions that are essential for H3K27me3 targeting.

Although LTRs without internal regions (solo-LTRs) exist, there are almost no ERVs with only internal sequences and no LTRs. We have now performed the suggested analysis of full length ERVs versus solitary LTRs, as summarized in the result section and illustrated by new screenshots in Figure 4—figure supplement 3. Solo-LTRs are present in staggering numbers in the genome and are most of the times interspersed with other repeats. We were therefore unable to analyze families of solo-LTRs collectively, since their epigenetic signatures are highly diverse, as they reflect the pattern of surrounding regions. Nevertheless, by focusing specifically on isolated solo-LTRs (not interspersed with other repeats), these were found to generally lack H3K9 or H3K27 methylation: this raises the exciting possibility that internal sequences are necessary for H3K9me3 or H3K27me3 recruitment for most ERVs. Interestingly, this was not observed for IAP elements, for which we indistinctly observed a high level of H3K9me3 enrichment at both full-length and solo-LTR copies.

*4) Strangely absent from the experiments/discussion about MERVL elements is the function of histone deacetylases (which can be inhibited pharmacologically with TSA) and LSD1 (KO ESC lines are available from multiple labs), which have both been shown to play an important role in MERVL silencing in ESCs (Macfarlan and Pfaff, Genes and Development 2011). Are these factors required for the re-silencing following the rapid DNA methylation?*

To address the referee’s request, we treated ES cells with TSA for 24h, either in serum or in 2i+vitC conditions. Our results show that MERVL dependency on HDACs is conserved, both in presence and absence of DNA methylation. This is now briefly commented in the Results section and appears in Figure 6—figure supplement 2.

*5) The experiment treating Dnmt 3KO ESCs with 2i and vit C is a nice experiment and important control. However the results shown in the supplementary figure are not that convincing, since the J1 controls look nothing like the ESCs used in the main figures. There are no statistics performed on these figures which is a bit concerning for the overall interpretation of the methodology.*

The data look different between the primary and supplementary figures because different techniques were used: RT-qPCR in primary, Nanostring in supplementary. Moreover, five biological replicates were used in primary, and eight in supplementary. As requested, we have now added statistical tests. In order to account for the different number of replicates (and variances) between WT and our various mutants, we used Welch’s t-test (adapted for unequal variances between conditions).

*6) I disagree with the assertion (made several times) that MERVL elements "switch from H3K9me3 to H3K27me3-regulated." MERVL elements are mostly devoid of H3K9me3 but instead have high levels of H3K9me2, which is lost upon LSD1 or G9a deletion (Macfarlan and Pfaff, G and D 2011, Maksakova and Lorincz, Epigenetics and chromatin 2013, also Figure 4). This is consistent with MERVLs being unresponsive to SETDB1 deletion. Thus this third category of transposons should be changed to reflect the data presented in the manuscript and the previous publications.*

We agree that MERVLs do not respond to KAP1 or ESET deletion: this has been shown by numerous studies, including in this manuscript. This excludes indeed a dependency towards ESET-dependent H3K9 trimethylation marks. However, our findings argue that H3K9me3 cannot be completely excluded as an actor of MERVL control. We found MERVLs to be strongly up-regulated in the absence of SUV39h enzymes, which are also H3K9 tri-methyltransferases. It is interesting to stress that this genetic link was previously found by an independent study (Bulut-Karslioglu 2014); in a supplementary figure, MERVLs appear as the most highly upregulated transposons in *Suv39h* mutants, but this result was not discussed. These data indicate that MERVL silencing relies on SUV39h in serum conditions.

Furthermore, we have now derived and analyzed triple-KO ES cells, which lack SUV39h and EED enzymes (Figure 6). This was instrumental in further demonstrating that the link between SUV39h and MERVL repression is probably direct. In serum-grown ES cells, while deletion of polycomb alone has no effect on MERVL, abolition of polycomb function in the *Suv39h*-mutant background resulted in even stronger MERVL activation compared to single *Suv39h* mutants. This indicates that polycomb-dependent control of MERVLs was likely implemented in serum-grown conditions, as a result of deficient SUV39h-dependent control.

MERVL-dependency towards SUV39h could be an indirect effect on G9a-dependent H3K9me2 deposition at transposons, as it has been shown that in absence of SUV39h, the G9a complex is destabilized, and reciprocally (Fritsch et al., 2010). However, this could also very likely reflect a genuine dependency towards SUV39h-related H3K9me3 marks.

Finally, even though the level of H3K9me3 enrichment is weak, it is not negligible, as we were able to detect this mark at MERVL sequences by ChIP. Based on these ChIP results and aforementioned genetic evidence, we strongly believe that although H3K9me2 is a strong determinant of MERVL repression in presence of DNA methylation, SUV39h-dependent H3K9me3 marks are also involved. For these reasons, we have now corrected the text and incorporated changes that highlight that MERVLs switch from H3K9me2 but also H3K9me3 to H3K27me3 chromatin.

*7) At day 6, IAP elements are derepressed but are still somewhat methylated. Can the authors address the methylation status of the expressed IAPs? This could perhaps be achieved by ChIP'ing first with H3K9me3, or with active marks like RNAPII or AcH3 marks, followed by bisulfite sequencing analysis.*

We thank the referee for suggesting this experiment. We have now performed H3K4me3- (to pick active elements) and H3K9me3- (inactive ones) ChIP followed by bisulfite pyrosequencing-based analysis of transposon methylation. The results are presented in Figure 4—figure supplement 2.

IAP elements indistinctly presented with the same DNA methylation level, whether they were marked by active or repressive marks. In contrast, LINE1 elements marked by H3K4me3 (and therefore potentially active) were clearly less DNA methylated that those marked by H3K9me3 (inactive ones). This experiment allowed us to further document the inter- and intra-familial heterogeneity of transposon regulation.

Reviewer #3:

*[…] However, a major pitfall of the current study is that all the analyses (DNA methylation, gene expression and histone modifications) are done at the cell population level during the time course of 13 days of 2i+vitC treatment. The authors have to consider the issue of cell heterogeneity in mid of 2i+vitC treatment (e.g. D6), which may lead to the different interpretations of their data. For example, it has been demonstrated that DNA hypomethylation-mediated de-repression of retrotransposons is associated with the degree of cell differentiation state (e.g. Hutnick et al. PMID 20404320, to be cited in this manuscript). When ESCs are grown in serum conditions, the 2i+vitC treatment would impose a selection, leading to the transient differentiation of a subset of ESCs in the time course.*

The question of cell heterogeneity is a very interesting one. We think it is unlikely that conversion from serum to 2i+vitC would induce a transient differentiation of a subpopulation of cells: 2i-based media rather induce the stabilization of pluripotency-associated transcription profiles and promote the acquisition of an epigenome closer to the one of the pluripotent inner cell mass of the blastocyst than serum-based conditions do (Ying et al., Nature 2008; Ficz et al., Cell Stem Cell 2013; Habibi et al., Cell Stem Cell 2013; Singer et al., Mol Cell 2014). The Hutnick et al. work was based on serum-grown conditions, in cells that were moreover genetically deficient for the DNA methylation machinery, which is not the case in our study.

*Therefore, as recently done by Grow EJ et al. (Nature 2015), the authors should perform immunocytochemistry or FISH analysis of ESCs to rule out the burst of transient de-repression of retrotransposons is only taking place in a subset of differentiated cells after 6days of 2i+vitC treatment.*

It seems that this referee missed the immunostaining analysis of transposon-encoded proteins (LINE1-ORF1 and IAP-Gag) we had performed in our original manuscript. It was said in the Results section: “While IAP-gag staining was uniform among cells at a given time point, LINE1 protein intensity showed great inter-cellular variability, ranging from intense to no signal, in both in serum (D0) and 2i+vitC conditions (D6 and D12)”. So indeed, we noticed some level of cellular heterogeneity in LINE1 expression among cells, but the same level of inter-cellular heterogeneity is present throughout the conversion process from D0 to D12, and, most importantly, is not different at the time of transcription burst (D6). In summary, what changes at D6 is the level of expression (as assessed by immunofluorescence intensity), not the number of expressing cells. This is now more clearly shown in Figure 2—figure supplement 2.

Moreover, to correlate the level of transposon expression with fluctuations in cellular states, we had performed co-staining of LINE1 and IAP with NANOG. We could not detect any correlation between NANOG and either LINE1 and IAP expression: high transposon levels could be indistinctly observed in cells with high or low NANOG staining, and reciprocally. This is mentioned in the Results section and appears as Figure 2—figure supplement 2.

In conclusion, we could not detect any association between pluripotency levels and transposon expression, and heterogeneity of LINE1 expression was present throughout the conversion. This suggests that the burst of expression is unlikely to be caused by a subpopulation of differentiated cells.

*Alternatively, multiplex single cell RT-PCR assays of ESCs with transposons and cell differentiation markers may also resolve this critical issue.*

Ideally indeed, single cell RT-qPCR analysis should be performed. But we would like to stress that single cell RT-PCR protocols are not adapted to the study of transposons. Considering the number of transposon copies in genomic DNA and the impossibility to design cDNA-specific primers due to their intron-less nature, DNase treatment is absolutely required when analyzing transposon transcripts by RT-PCR-based methods. Single cell RT-qPCR protocols, as the ones developed with the Fluidigm technology, do not include a DNase treatment: the results would therefore be completely unreliable.

*1) In the section of "Transposons undergo a biphasic mode of regulation upon serum to 2i+vitC conversion". This section needs to be resolved by IHC or multiplex single cell RT-PCR to determine whether the bursting of retrotransposon expression is only in a subset of heterogeneous cells during the time course of 2i+vitC conversion.*

See answer above: we had already performed immunofluorescence-based cellular detection of transposon-encoded proteins and pluripotency markers in our original submission.

*2) In Figure 3, the authors showed RNA-seq analysis of differential gene expression during the 2i+vitC conversion. It would be obvious to the authors whether they have detected the up-regulation of differentiation genes such as ectoderm, mesoderm and endoderm marker genes in D6 samples. In fact, the transient upregulation of Tbx3 gene is consistent with this possibility (Figure 3).*

At the beginning and during the conversion from serum to 2i+vitC, all the ectoderm/mesoderm/endoderm marker genes we could look at remained very lowly expressed.

*3) Statistical analysis should be performed for many data points. For example, in Figure 6, are the differences of expression in different time points statistically significant?*

As requested, statistical analyses have been systematically added now, using Welch’s t-test (unequal variances t-test).

*4) In Figure 2, it is mentioned that Dnmt-tKO cells did not show transposon transcriptional reactivation, which is in strong contrast to the 2i+vitC. The explanation given is that long-term compensatory mechanisms have repressed these sites. However, there is no established mechanism. If H3K9me3 is not increased in Dnmt-tKO, does H3K27me3 increase in Dnmt-tKO under 2i-vitC conditions? It is worthwhile to study or discuss the discrepancy in more detail.*

We showed in our original submission (based on available H3K27me3 ChIP-seq datasets) that *Dnmt-*tKO cells grown in serum have established in the long term H3K27me3-dependent control of transposons, similarly to ES cells after more than 6 days of 2i+vitC conversion: they also have increased H3K27me3 at transposons and the distribution of this mark along the sequence of the different transposon categories is also the same. These results are presented in Figure 5—figure supplement 2. We hope that the referee will agree that there is no point studying *Dnmt-tKO* cells in 2i+vitC conditions as these features are already present in serum. This highlights that H3K27me3 gain is not linked to the change of culture condition but is an adaptive response to the hypomethylated state of the cells, therefore providing a compensatory mechanism for DNA methylation loss.